# From Sequence to Structure: Uncovering Substructure Reasoning in Transformers

**Xinnan Dai**[1][†][*], **Kai Yang**[2][†], **Jay Revolinsky**[1], **Kai Guo**[1][*],
**Aoran Wang**[3], **Bohang Zhang**[2], **Jiliang Tang**[1]
[1]Michigan State University, [2]Peking University, [3]University of Luxembourg
{daixinna, revolins, guokai1, tangjili}@msu.edu, yangkai@alumni.pku.edu.cn,
zhangbohang@pku.edu.cn, ralf.wong@outlook.com

## Abstract

Recent studies suggest that large language models (LLMs) possess the capability to solve graph reasoning tasks. Notably, even when graph structures are embedded within textual descriptions, LLMs can still effectively answer related questions. This raises a fundamental question: *How can a decoder-only Transformer architecture understand underlying graph structures?* To address this, we start with the substructure extraction task, interpreting the inner mechanisms inside the transformers and analyzing the impact of the input queries. Specifically, through both empirical results and theoretical analysis, we present Induced Substructure Filtration (ISF), a perspective that captures the substructure identification in the multi-layer transformers. We further validate the ISF process in LLMs, revealing consistent internal dynamics across layers. Building on these insights, we explore the broader capabilities of Transformers in handling diverse graph types. Specifically, we introduce the concept of thinking in substructures to efficiently extract complex composite patterns, and demonstrate that decoder-only Transformers can successfully extract substructures from attributed graphs, such as molecular graphs. Together, our findings offer a new insight on how sequence-based Transformers perform the substructure extraction task over graph data.

## 1 Introduction

It is evident from recent studies that large language models (LLMs) are capable of understanding structured data [31, 22, 15]. For example, when graph structures are presented in textual sequence, LLMs can identify node connections [10, 21], detect graph patterns [3, 7], and compare common subgraphs across a given set [20, 7]. However, transformers, which serve as the backbone of LLMs, are inherently designed for sequential textual data, which does not naturally capture graph structures. This gap raises a fundamental question: How can a sequence-based decoder-only transformer comprehend structured data like graphs?

To answer this question, existing research focuses mainly on basic graph reasoning tasks to build the concept of the mechanism by which transformers understand graph structures [17, 29]. The shortest path is one of the basic tasks [21, 6, 1]. Based on the shortest path task, SLN [5] suggests that a form of spectral navigation implicitly emerges within Transformer layers, enabling global coordination across nodes. Meanwhile, Abulhair et al. [18] and ALPINE [25] argue that Transformers learn to find paths by composing and merging multiple candidate paths based on the provided edge list. However, these studies are limited to linear paths, while real-world graphs often contain more complex, non-linear substructures such as cycles, trees, and other motifs. As a result, existing understandings drawn

---

[*]corresponding: {guokai1, daixinna}@msu.edu, † equal contribution, code is available at
`https://github.com/DDigimon/From_Sequence_to_Structure`

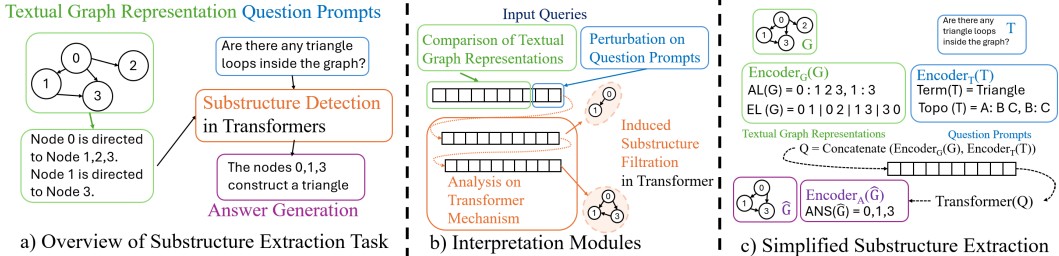

Figure 1: An overview of the interpretation for the substructure extraction task. a) Substructure Extraction Task: The Transformer receives a graph description and a question prompt as input and generates an answer. b) Interpretation Modules: The analysis includes input queries and internal Transformer processing. c) Simplified Substructure Extraction: The extraction process is simplified to highlight the core mechanism.

from path-finding tasks may not generalize to comprehensive graph understanding and are limited to explain why LLMs can do various graph tasks.

In this work, we explore how Transformers tackle the broader challenge of substructure understanding, with a particular emphasis on the task of substructure extraction. Building on the use of LLMs for substructure extraction [7, 3], Figure 1a) illustrates the overall process with Transformers: Transformer-based models receive a query prompt, which is composed of a textual graph representation and a question prompt as input. Then, they identify the relevant substructure, and generates an answer for the given graph.

To investigate how Transformers derive answers from input tokens, in Section 3 we conduct empirical and theoretical analyses of train-from-scratch Transformers, focusing on the internal Transformer mechanisms and input queries, as shown in Figure 1b). To understand the internal behavior of the model, we introduce a new perspective, Induced Substructure Filtration (ISF), which suggests that Transformers perform a layer-wise node aggregation process to detect substructures.

To verify the reliability of our interpretation modules, we demonstrate that our approach also applies to understanding LLM behavior and argue that it can inform the development of future methods. In Section 4, we show that our explanation for Transformers aligns with the behaviors observed in LLMs, particularly in how they tackle various textual graph representations and perform graph extraction tasks. Furthermore, in Section 5, we explore the potential of Transformers in graph understanding, building on insights from our interpretation modules. We argue that it is reasonable to extend Transformers to handle attributed graphs, such as molecular graphs. Finally, we introduce the Thinking-In-Substructure framework, which enhances Transformers' capabilities in complex graph reasoning tasks. In summary, we offer a new perspective on how Transformers understand graph structures from sequential inputs. Our key contributions are summarized as follows:

1. We provide insights into how Transformers extract substructures, based on experiments and theory, focusing on internal mechanisms and input queries.

2. We propose Induced Substructure Filtration (ISF) to explain how Transformers identify substructures across layers.

3. We show that our interpretation is applicable to LLMs, explaining their behaviors in graph tasks and supporting the extension of Transformers to attributed graphs and more complex graph reasoning.

## 2 Preliminary

We combine theoretical insights and experimental results to demonstrate how Transformers solve the substructure extraction task. To support this, we briefly introduce the core notations and definitions used in this work and provide an overview of the experimental setup for the following empirical studies.

## 2.1 Problem formulations

**Substructure extraction in transformers**  Although we prompt LLMs to interpret natural language sentences to answer substructure-related questions, we simplify the process by converting these sentences into symbolic tokens. This enables us to analyze and train Transformers from scratch. The simplified process is illustrated in Figure 1 c). Given a graph $G = \{V, E\}$ and a question prompt $T$, we encode them using $\text{Encoder}_\text{G}$ and $\text{Encoder}_\text{T}$ respectively to obtain simplified sentence sequences. These are then concatenated, with the encoded question placed after the graph representation, forming the input query $Q$. The objective is to extract the set of isomorphic subgraphs $\hat{G} = \{g_1, g_2, \cdots, g_s\}$, where each $g_i$ represents an instance of the desired substructure within the input graph. The overall process is defined as: $\text{Encoder}_\text{A}(\hat{G}) = \text{Transformers}(Q) = \text{Transformers}(\text{Encoder}_\text{G}(G), \text{Encoder}_\text{T}(T))$. We introduce each encoder of this framework in the following paragraphs.

**Textual graph representations** ($\text{Encoder}_\text{G}$)  To input graphs into a Transformer, prior work [10, 7] often converts them into textual sequences using either the Adjacency List (AL) or Edge List (EL). Specifically, for a graph $G = \{V, E\}$ and a vertex $v_i \in V$, the AL format captures its neighborhood $N(v_i) = \{v \in V \mid (v_i, v) \in E\} = \{v_i^1, \cdots, v_i^{m_i}\}$, where $m_i$ is the number of neighbors of nodes $v_i$. In the textual representation, each node and its neighbors are formatted as a sentence: the central node and its neighbors are separated by a colon ":", and different such groups are separated by commas ",", which formulated as:

$$\text{AL}(G) = (v_1; \text{``:''}; v_1^1; \cdots; v_1^{m_1}; \text{``,''}; \cdots; \text{``,''}; v_n; \text{``:''}; v_n^1; \cdots; v_n^{m_n}).$$

Instead of focusing on central nodes, the EL format enumerates all possible edges $(v_i, v_j) \in E$. Each edge pair is separated by a vertical bar "|". The representation of EL is formulated as:

$$\text{EL}(G) = (v_1; v_1^1; \text{``|''}; \cdots; \text{``|''}; v_1; v_1^{m_1}; \text{``|''}; \cdots; \text{``|''}; v_n; v_n^1; \text{``|''}; \cdots; \text{``|''}; v_n; v_n^{m_n}).$$

The details of the definitions are in the Definition D.1 and Definition D.2 in Appendix D.1.

**Question prompt** ($\text{Encoder}_\text{T}$)  Next, we define the question prompt to determine which substructures should be extracted from the input graph. This prompt, denoted as instruction $T$, can be either terminology-based or topology-based, as described in [7]. If the substructures are well-known, such as a "triangle", they can be defined using either terminology or topological instructions, represented as $\text{Term}(T) = (\text{triangle})$ and $\text{Topo}(T) = (A : BC, B : C)$, respectively. However, in most cases, the substructures are not clearly defined by terminology, so we rely on topology-based definitions.

**Answer generation** ($\text{Encoder}_\text{A}$)  The output of the Transformer is a text sequence. However, this sequence must correspond to a unique substructure. To align the substructures with the text output, we constrain the Transformer to output the node sets for each substructure, separated by commas. Formally, the output is represented as: $\text{ANS}(\hat{G}) = (v_1^{g_1}, v_2^{g_1}, \ldots, v_w^{g_1}, \text{``,''}, \ldots, \text{``,''} v_1^{g_s}, v_2^{g_s}, \ldots, v_w^{g_s})$, where each group $\{v_1^{g_i}, \ldots, v_w^{g_i}\}$ denotes the nodes in subgraph $g_i$, and commas "," are used to delimit different substructures.

## 2.2 Experiment settings

**Transformer training**  We train Transformer models using the same architecture as GPT-2 but in a lightweight version, with only 384 hidden dimensions and a small number of layers depending on the tasks. The details for each task are shown in the Appendix E. During training, the model is optimized only to predict $\text{ANS}(\hat{G})$. For evaluation, we use accuracy as the metric. A predicted answer is considered correct only if the $\text{ANS}(\hat{G})$ is exactly the same with the ground truth.

**Dataset setting**  We generate over 5 million directed graphs, with node counts ranging from 4 to 16 and edge counts from 3 to 120. The graphs are constructed based on specific requirements detailed in the following empirical studies. To prevent result copying from the same graphs, we ensure that the graphs in the training and testing sets are non-isomorphic.

Table 1: Transformers extract the substructures from the given graph sequence

| # Training | # Layer | Triangle | Path | Square | Diagonal | T_triangle | F_Triangle | Diamond | Pentagon | House |
|---|---|---|---|---|---|---|---|---|---|---|
| 100K | 2 | $0.5301 \pm 0.06$ | $0.5534 \pm 0.03$ | $0.1936 \pm 0.04$ | $0.1163 \pm 0.00$ | $0.1911 \pm 0.03$ | $0.2877 \pm 0.03$ | $0.0656 \pm 0.01$ | $0.3628 \pm 0.01$ | $0.3705 \pm 0.00$ |
|  | 3 | $0.9662 \pm 0.00$ | $0.8066 \pm 0.02$ | $0.3991 \pm 0.00$ | $0.4417 \pm 0.07$ | $0.4974 \pm 0.02$ | $0.5329 \pm 0.04$ | $0.1635 \pm 0.03$ | $0.5638 \pm 0.01$ | $0.5603 \pm 0.00$ |
| 300K | 3 | $0.9948 \pm 0.00$ | $0.9195 \pm 0.01$ | $0.7247 \pm 0.01$ | $0.6313 \pm 0.07$ | $0.7775 \pm 0.02$ | $0.7831 \pm 0.06$ | $0.6189 \pm 0.02$ | $0.7063 \pm 0.05$ | $0.7455 \pm 0.03$ |
|  | 4 | $0.9947 \pm 0.00$ | $0.9493 \pm 0.02$ | $0.9403 \pm 0.02$ | $0.9140 \pm 0.02$ | $0.9080 \pm 0.03$ | $0.8097 \pm 0.02$ | $0.8765 \pm 0.02$ | $0.8634 \pm 0.00$ | $0.8386 \pm 0.04$ |
| 400K | 4 | $0.9802 \pm 0.02$ | $0.9802 \pm 0.01$ | $0.9620 \pm 0.00$ | $0.9596 \pm 0.00$ | $0.9287 \pm 0.02$ | $0.8534 \pm 0.01$ | $0.9048 \pm 0.03$ | $0.8612 \pm 0.01$ | $0.8023 \pm 0.05$ |
|  | 5 | $0.9977 \pm 0.00$ | $0.9948 \pm 0.00$ | $0.9679 \pm 0.02$ | $0.9707 \pm 0.01$ | $0.9430 \pm 0.04$ | $0.8750 \pm 0.02$ | $0.9306 \pm 0.02$ | $0.8922 \pm 0.01$ | $0.8530 \pm 0.02$ |

# 3 Interpretations for Substructure Extraction in Transformers

In this section, we present our insights into how Transformers perform substructure understanding. We focus on two main aspects: the internal mechanisms of Transformers in solving the substructure extraction task, discussed in Section 3.1, and the impact of input query formulation on extraction performance Section 3.2.

## 3.1 Induced Substructure Filtration in Transformer

In this subsection, we introduce how Transformers solve the substructure extraction task. First, we show that Transformers can extract substructures of diverse shapes, as detailed in Section 3.1.1. We then analyze the underlying mechanism and propose the ISF process in Section 3.1.2. Finally, we demonstrate how ISF generalizes to cases with multiple substructures of varying numbers and shapes in Section 3.1.3.

### 3.1.1 Single substructure extraction

We begin with the Single-Shape-Single-Num case, evaluating whether Transformers can extract a specific target substructure from a given graph whose scale and shape may vary. The selected substructures contain 3 to 5 nodes and 3 to 6 edges. We also investigate the effects of dataset size and the number of Transformer layers. To this end, we vary the training set size from 100K to 400K and the number of Transformer layers from 2 to 5. For each substructure, we evaluate the extraction accuracy on 30K test graphs, averaging results over three runs. Table 1 shows the results for various substructure extraction tasks.

The Transformers are capable to extract the target shape of substructures from the given graph with at least 2 layer transformers, achieving over 85% accuracy. However, different substructures exhibit varying requirements in terms of both data scale and model depth. For instance, the 3-cycle (triangle) structure can achieve 99% accuracy with just 3 layers and 100K training examples, while the 5-cycle (pentagon) structure requires over 5 layers and at least 400K examples to attain comparable performance. Furthermore, we observe that the minimum number of Transformer layers required to achieve 85% accuracy correlates with the number of nodes in the substructure. For example, all 4-node substructures can achieve promised results with 4-layer transformers, while 5-node substructures need 5-layer transformers. This suggests that the number of layers is a crucial factor in a Transformer's ability to understand graph structures. To understand how the number of layers affects a Transformer's grasp of graph structures, we analyze substructure extraction across layers in the following subsections.

### 3.1.2 Induced Subgraph Filtration

**Visualization Results** We visualize token embeddings to better understand how Transformers extract substructures. Since decoder-only Transformers process input left to right [2], we use the final token embeddings to reflect their graph understanding. We apply t-SNE to project these embeddings into a 2D space, labeling each graph by its substructure answer, which is represented by the node IDs as illustrated in $\mathsf{ANS}(\hat{G})$. As an example, we use the square substructure extraction task, shown in Figure 2. The legends are the node IDs. For example, "0431" indicates that the model first identifies node 0 (with out-degree 2), followed by its neighbor 4 (out-degree 1), then node 3 (a neighbor of 4), and finally node 1 (a neighbor of both 3 and 0)

The visualization results reveal how Transformers identify substructures across layers, with graphs sharing similar answers gradually clustering together. Although the visualization targets the final graph token, the substructure answers are already determined by the last layer before the generation

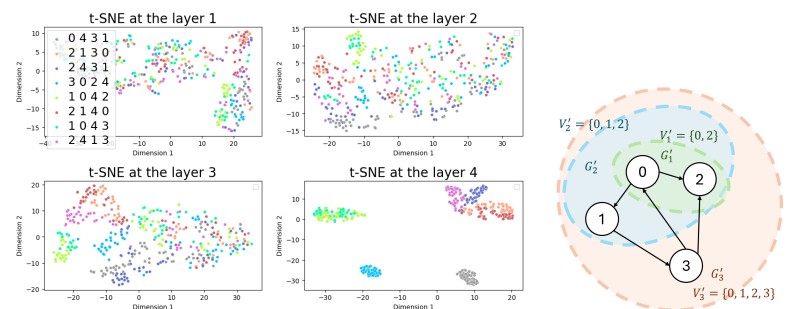

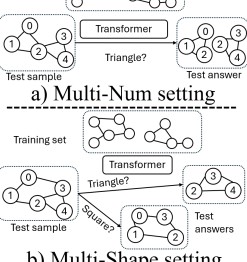

Figure 2: Visualization across 4 layers. We show the node ID distributions of target substructures. Legends indicate the node IDs

Figure 3: 4-Node 3-Filtration and Induced Subgraph Filtration

Figure 4: Tasks in Simultaneous detection

step. We also observe that substructures form progressively across layers. For example, in layer 2, graphs with substructure ID '2431' (dark blue) begin clustering near '0431' (grey) and '2413' (purple), sharing '31' and '24'. By layer 3, '2431' moves closer to '0431', which shares '431'. In the final layer, substructure types are clearly separated. Since outputs follow a left-to-right order, those with similar starting tokens cluster together. Transformers infer substructures before generation, progressively organizing substructures across layers.

**Theoretical modeling** We further provide a theoretical framework for this process by introducing filtrations to formalize progressive substructure extraction. Using the square extraction visualization as an example, we model it as a 4-node, 3-filtration process, indicating that the target substructure contains 4 nodes and requires 3 filtration steps, each corresponding to an induced subgraph. More generally, we define this framework as a $k$-Node $m$-Filtration, referred to as Induced Subgraph Filtration, as defined in Definition 3.1.

**Definition 3.1** ($k$-Node $m$-Filtration and Induced Subgraph Filtration). A *$k$-node $m$-filtration* on $V'$ ($|V'| = k$) is $\mathcal{F}(V') = (V_1', \ldots, V_m')$ where $\varnothing \neq V_1' \subseteq \cdots \subseteq V_m' = V'$. For $G' = (V', E')$, this yields an *induced subgraph filtration* $(G_1', \ldots, G_m')$ where $G_i' = G'[V_i']$.

Figure 3 illustrates the concept defined in Definition 3.1. We model the extracted substructure as $G' = (V', E')$ with $|V'| = k$, and represent the extraction process using an $m$-Filtration, where $m$ denotes the number of gathering operations required for the Transformers to identify the substructure.

Further, to capture the matches of $G'$ in $G = (V, E)$, we define a Substructure Isomorphism Indicator Tensor.

**Definition 3.2** (Subgraph Isomorphism Indicator Tensor). For graphs $G = (V, E)$ ($|V| = n$) and $G' = (V', E')$ ($|V'| = k$), the *subgraph isomorphism indicator tensor* $\mathcal{T}(G, G')$ is $k$-dimensional ($n \times \cdots \times n$) where its entry for an ordered $k$-tuple of vertices $(v_{j_1}, \ldots, v_{j_k})$ from $V$ is 1 if these vertices induce a subgraph isomorphic to $G'$ (via a predefined mapping $v_p' \mapsto v_{j_p}$ for $p = 1, \ldots, k$), and $\mathcal{T}_{j_1, \ldots, j_k} \leq 0$ otherwise (see Definition D.5 for details).

Theorem 3.3 (proof in Appendix D.3) shows that Transformers can progressively compute $\mathcal{T}(G, G')$ for each substructure along the filtration. $O(n^k)$ is the hidden dimension needed for the Transformer to check all $k$-node subgraphs in an $n$-node graph.

**Theorem 3.3** (Expressiveness for Progressive Identification). *Given a $k$-node $m$-filtration $\mathcal{F}(V')$ on $V' = \{v_1', \ldots, v_k'\}$. For any directed graphs $G = (V, E)$ ($|V| = n$) and $G' = (V', E')$, a log-precision Transformer with $m + 2$ layers, constant heads, and $O(n^k)$ hidden dimension can output $\mathsf{vec}(\mathcal{T}(G, G'[V_i']))$ at layer $i + 2$ for $i \in \{1, \ldots, m\}$.*

Furthermore, the Transformer extracts the unique instance of $G'$, meaning the answer is uniquely determined by the given graph representation and question prompt, and $\mathcal{T}(G, G')$ contains exactly one entry equal to 1. This leads to Assumption 3.4 and Theorem 3.5. With this condition, the substructure extraction task is solvable for transformers.

**Assumption 3.4** (Single-Shape-Single-Num). For graphs $G, G'$, there is a *unique* $k$-tuple of indices $(i_1, \ldots, i_k)$ for which $\mathcal{T}(G, G')_{i_1, \ldots, i_k} = 1$.

**Theorem 3.5** (Expressiveness for Pattern Extraction). *Under Assumption 3.4, for directed graphs $G = (V, E)$ ($|V| = n$) and $G' = (V', E')$ ($|V'| = k$), a log-precision Transformer with constant depth, constant heads, and $O(n^k)$ hidden dimension can output the unique $k$-tuple of vertices $(v_{i_1}, \ldots, v_{i_k})$ for which $\mathcal{T}(G, G')_{i_1, \ldots, i_k} = 1$.*

*Remark* 3.6. Theorems 3.3 and 3.5 hold for various input graph representations (e.g., adjacency lists $\mathsf{AL}(G)$ or edge lists $\mathsf{EL}(G)$), as formally defined in Definitions D.1 and D.2.

### 3.1.3 Simultaneous detection of multiple substructures

As graphs often contain multiple and diverse substructures, we further demonstrate how the ISF process adapts to such scenarios. Specifically, we evaluate whether a 4-layer Transformer can accurately detect both repeated and differently shaped substructures. To this end, we design the Single-Shape-Multi-Num and Multi-Shape-Single-Num evaluation settings, with the pipeline illustrated in Figure 4.

**Single-Shape-Multi-Num** As shown in Figure 4 a), the Single-Shape-Multi-Num task involves training and testing on graph sets where each sample may contain multiple target substructures—up to five in total. This task evaluates whether Transformers can successfully extract all target substructures within a single graph. We define the task in Definition 3.7.

**Definition 3.7** (Single-Shape-Multi-Num Extraction). The Single-Shape-Multi-Num extraction task requires a model to output all $k$-tuples of vertices $(v_{i_1}, \ldots, v_{i_k})$ corresponding to occurrences of directed graph $G' = (V', E')$ ($|V'| = k$) in $G = (V, E)$ ($|V| = n$). Formally, the objective is to output all tuples satisfying $\mathcal{T}(G, G')_{i_1, \ldots, i_k} = 1$, where $\mathcal{T}(G, G')$ is Subgraph Isomorphism Indicator Tensor defined in Definition 3.2.

As shown in Figure 5, Both triangles and square detections can achieve over 85% accuracy. The number of substructures has little impact on the Transformers' ability. They can identify multiple patterns at once. We further analyze examples with two substructures and find that answers are still often determined before the final generation step (Figure 7). The theoretical explanation is provided in Theorem 3.8.

**Theorem 3.8** (Expressiveness for Single-Shape-Multi-Num Extraction). *Fix integers $n \geq k \geq 1$. There exists a log-precision Transformer with constant depth, constant number of attention heads, and $O(n^k)$ hidden dimension that can complete Single-Shape-Multi-Num Extraction defined in Definition 3.7 for directed graphs $G = (V, E)$ ($|V| = n$) and $G' = (V', E')$ ($|V'| = k$).*

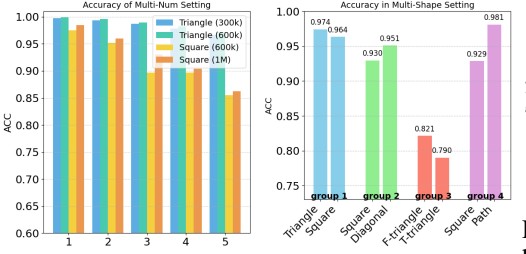
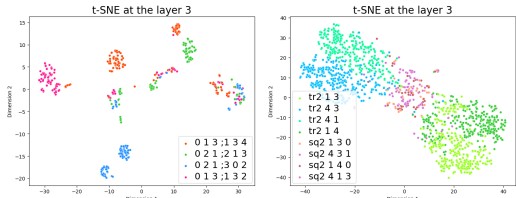

Figure 5: The Multi-Num setting results

Figure 6: The Multi-Shape setting results

Figure 7: Transformers has organized the answers before the generate the answers.

Figure 8: Triangles (tr) are identified at layer 3 when trained with squares (sq).

**Multi-Shape-Single-Num** As illustrated in Figure 4 b), the Multi-Shape-Single-Num task involves training and testing on graph sets where each sample may contain multiple substructures of different shapes. This task evaluates whether Transformers can identify diverse substructure types within a single graph defined in Definition 3.9.

**Definition 3.9** (Multi-Shape-Single-Num Extraction). The Multi-Shape-Single-Num extraction task requires a model to find all the occurrences for any directed graph $G' = (V', E')$ ($|V'| \leq k$) in $G = (V, E)$ ($|V| = n$) satisfying Assumption 3.4. Formally, the objective is to output the *unique* $k'$-tuple of vertices $(v_{i_1}, \ldots, v_{i_{k'}})$ for which $\mathcal{T}(G, G')_{i_1, \ldots, i_{k'}} = 1$, where $\mathcal{T}(G, G')$ is Subgraph Isomorphism Indicator Tensor defined in Definition 3.2.

We create this task by mixing single-shape training data and dividing it into four groups (details in Appendix E.2), as shown in Figure 6. Each substructure can be extracted independently, guided by its specific question prompt. Therefore, in visualization, we take the embedding of the last input query token rather than the last graph representation token. From Figure 8, we find that Transformers often identify simpler substructures, like triangles, in layer 3, instead of delaying all predictions to the final layer (the Transformer has 4 layers in this setting). We summarize the mechanism in Theorem 3.10.

**Theorem 3.10** (Expressiveness for Multi-Shape-Single-Num Extraction). *Fix integers $n \geq k \geq 1$. There exists a log-precision Transformer with constant depth, constant heads, and $O(n^k)$ hidden dimension that can complete Multi-Shape-Single-Num Extraction defined in Definition 3.9 for a directed graph $G = (V, E)$ ($|V| = n$) and any target subgraph $G' = (V', E')$ with $|V'| = k' \leq k$ satisfying Assumption 3.4.*

These two properties form the foundation for understanding how decoder-only Transformers decompose complex graphs into simpler ones for substructure extraction, as discussed in Section 5.1.

### 3.2 Impact of Input Query Formulation

As illustrated in Section 2, the input query consists of two components: a text-based graph representation and a question prompt. In Section 3.2.1 and Section 3.2.2, we discuss how Transformers perform the substructure extraction task based on these inputs.

#### 3.2.1 Text-Based Graph Representations

We start with the comparison of different text-based graph representation methods. Specifically, we focus on the two basic methods, which are the neighborhood-based AL and the edge-based EL. To ensure controllable input lengths, we conduct a toy experiment using graphs with 4 to 8 nodes, keeping both representations within a 100-token limit. We then vary the number of Transformer layers to extract substructures (e.g., triangle, square, or pentagon) from the graph representations. Experimental details are provided in the Table 7 in Appendix E.1 and the results are shown in Figure 9.

The experimental results indicate that both AL and EL formats allow Transformers to extract substructures from text-based graph representations. As the size of the target substructure increases, both formats require more Transformer layers to achieve more than 80% accuracy. As mentioned in Remark 3.6, the theoretical results hold for both AL and EL formats. The intuition is that both formats can be transformed into the same binary adjacency matrix $A(G)$, which encodes the structure of graph $G$. This matrix is then vectorized as $\text{vec}(A(G))$ for processing within the model, where we provides the details in Lemma D.6 in Appendix D.2.

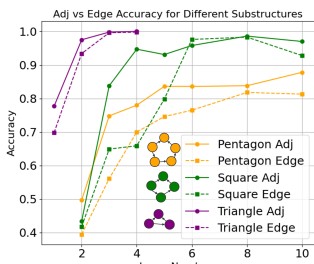

Figure 9: Using AL and EL to predict substructures with varying numbers of Transformer layers.

Although we theoretically show that EL and AL are equivalent in representational power, experimental results reveal that EL requires more Transformer layers to achieve comparable performance. This may be because EL inherently requires more tokens to explicitly represent all edges, whereas AL encodes the same information with fewer tokens and additional padding, benefiting from its more compact structure. In practice, for a fully connected graph, the AL needs $2 \times |V|^2 - |V|$ tokens, while the EL needs $3 \times (|V|^2 - |V|)$ tokens. Therefore, for efficiency, we mainly adopt AL in our discussions. Appendix E.3 provides the details of the training efficiency comparison on AL and EL.

#### 3.2.2 Question prompt encoders

We explore how question prompts influence structural understanding in substructure extraction. The question prompt $T$ claims the target substructure, expressed as either terminology-based $\text{Term}(T)$ (e.g., "triangle") or topology-based $\text{Topo}(T)$. In $\text{Topo}(T)$, we use AL-style descriptions with symbolic node labels, e.g., a triangle as $\text{Topo}(T) = (A : BC, B : C)$.

We construct a mixed data set to train the Transformer, following the setup in Section 3.1.3, but balance it with equal samples using topology-based and terminology-based prompts, denoted "Term", "Topo1" and "Topo2" in Table 2. The "Term" uses semantic labels (e.g., "Triangle"), while "Topo1"

Table 2: Results on different question prompts. "p" means the pad token.

| Mixture training | Term | ACC | Topo1 | ACC | Topo2 | ACC | Symbol-level | ACC | Token-level | ACC |
|---|---|---|---|---|---|---|---|---|---|---|
| Group1 | Triangle | 0.9782 | A:BC,B:C | 0.9794 | B:AC,A:C | 0.9166 | :,: | 0.9152 | C/D | 0.7074 / 0.1027 |
|  | Square | 0.8478 | A:D,C:BA,D:B | 0.8494 | B:AD,A:C,C:D | 0.8500 | :,:,: | 0.7444 | C/D | 0.7532 / 0.8470 |
| Group2 | Diagonal | 0.9082 | A:BCD,C:D,D:B | 0.2332 | B:D,C:ABD,D:A | 0.7354 | p:pp,p:p,p:p | 0.7086 | A/C | 0.8566 / 0.2991 |
|  | Square | 0.8810 | A:BC,C:D,D:B | 0.8691 | B:AD,A:C,C:D | 0.9037 | p:p,p:ppp,p:p | 0.7106 | A/C | 0.1271 / 0.9094 |

gives direct node connections and "Topo2" describes the same structure with shuffled node names. To examine how Transformers align different descriptions with graph inputs, we introduce two types of perturbations: symbol-level and token-level, where these tokens are used directly as question prompts. Symbol-level perturbations test the impact of structural phrasing, while token-level perturbations assess reliance on specific tokens within topology prompts. Results are reported in Table 2. In token-level perturbation, we use two different tokens to examine whether they have distinct impacts on Transformer performance.

The results show that Transformers can use both terminology- and topology-based prompts, achieving over 70% accuracy in each case. However, terminology-based prompts perform better, reaching over 85% in Group 2. For example, Topology-based questions show limitations, struggling to represent diagonal substructures accurately. Perturbation results suggest that predictions often rely on specific symbolic cues or tokens rather than full structural understanding. For instance, in Group 1, triangles are identified via symbolic patterns, while in Group 2, diagonal and square structures are distinguished by tokens like "A" and "C.". This suggests that Transformers do not explicitly learn full structural representations or map them back to the original graphs for answers. Instead, they abstract substructure concepts using a series of key tokens.

## 4  Consistency in LLM Graph Understanding Behavior

As discussed in Section 3, we identify three key findings: In terms of the Transformer mechanism, (1) Transformers perform the ISF process to extract substructures simultaneously. Regarding input formulation, (2) both EL and AL can represent the adjacency matrix $A(G)$, though EL may perform slightly worse than AL due to sequence length limitations, and (3) Transformers tend to abstract substructures into a sequence of tokens in the question prompt, rather than fully capturing the underlying topological concepts. Since decoder-only Transformers are a common architecture in modern LLMs, we further evaluate whether LLMs exhibit the ISF process during substructure extraction, and whether our understanding of input formulation can explain their behavior.

**Induced Subgraph Filtration**  To investigate whether LLMs exhibit the ISF process, we visualize the fine-tuned LLaMA 3.1-8B-Instruct model on a triangle detection task (details in Appendix E.4.2), as shown in Figure 10. The results align with our findings in Section 3.1: the model often identifies the correct answer before generating it, and deeper layers better distinguish between similar answers. For example, the responses of the substructures in node ID '302' and '304' become more separable at layer 23 than at layer 17. We quantify this trend in Figure 11, where Adjusted Rand Index (ARI) and Normalized Mutual Information (NMI) scores increase with depth. However, unlike Transformers trained from scratch,

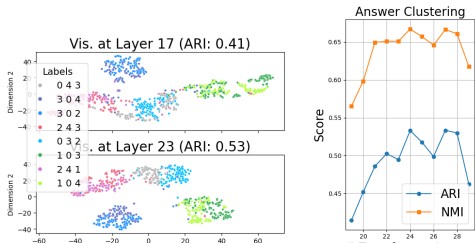

Figure 10: Visualization on Llama3.1-8B-Instruct

Figure 11: ARI and NMI across the layers.

which only predict answers, LLMs can generate explanatory content. For instance, 23% of responses include Python code, while others describe node relationships in the graph. As a result, ARI and NMI scores slightly decrease in the ending layers.

**Text-based Graph Representation**  Current evaluations have discussed the effect of how different graph representations influence the LLMs in graph reasoning tasks. For example, GraphMem [31] demonstrates that LLMs possess the ability to transfer knowledge across different graph descriptions. This is possible because both AL and EL can be mapped to $\text{vec}(A(G))$, making them share the representations at the graph representation level. However, since the EL typically requires more tokens to describe a graph and LLMs have limitations in handling long contexts, prior studies [10, 6] report that EL representations sometimes perform worse than AL descriptions.

**Question Prompt**   In the context of substructure understanding, GraphPatt [7] introduces both terminology-based and topology-based descriptions for the substructure extraction task, showing that terminology-based prompts generally lead to better performance. Moreover, a single terminology concept can correspond to multiple diverse topology-based descriptions reported in [7].

## 5    Understanding on Complex Graphs

In Section 3, we highlight the ISF process as a key mechanism enabling Transformers to solve the substructure extraction task. Moreover, we observe that when text-based descriptions can be mapped to the adjacency matrix $A(G)$, Transformers can perform substructure extraction on the given graph. Building on these insights, we explore broader applications: in Section 5.1, we introduce a new method for efficiently reasoning over composite substructures and in Section 5.2, we examine how Transformers adapt to attributed graphs.

### 5.1    Thinking in substructures

As introduced in Section 3, Transformers apply the ISF process to extract substructures. This process runs synchronously across different patterns, with smaller ones being easier to detect. Besides, complex structures often consist of simpler components, which we refer to as decomposing substructures. Building on these observations, we propose the Thinking-in-Substructure (Tins) method to explain how decoder-only Transformers solve complex substructure reasoning tasks. For example, [7] reports that reasoning language models decompose a house pattern into a triangle and a square to search for the substructures. We reformulate the answer generation part as $\mathsf{ANS}_{\mathsf{Tins}}(\hat{G}) = (\{P_1\}, \{P_2\}, \ldots, \{P_t\}, < \mathrm{ANS} >, \mathsf{ANS}(\hat{G}))$, where $\{P_i\}$ is the collection of each decomposing structure and $< \mathrm{ANS} >$ is a special token to indicate that the followings are final answers. This decomposition reduces the extraction complexity from $O(n^k)$ to $O(n^q)$, where $q$ and $k$ are the maximum size of decomposing substructures and target substructures, respectively, with $q < k$. We show the proof in Theorem D.13 in Appendix D.4.

To verify the efficiency of Tins, in the experiment, we design 4 different composite substructures with their decomposition process trained with 100K samples. The experiment settings are in Appendix E.4.3. The results suggest that Tins can help transformers significantly improve the performance with limited training data. The overall performance can increase 10%, and in the 3 layer for diagonal structure, the performance increases about 46% percent.

Table 3: The results of Thinking-in-substructures (Tins)

| Substructures | Directly Preds 4 layer | 3 layer | Decomposition | Tins 4 layer | 3 layer |
|---|---|---|---|---|---|
| Diagonal | 0.6314 | 0.1998 | | 0.8648 | 0.6606 |
| Diamond | 0.4756 | 0.1288 | | 0.7792 | 0.4338 |
| House | 0.5887 | 0.5640 | | 0.8066 | 0.6678 |
| Complex | 0.1182 | 0.1208 | | 0.2268 | 0.2124 |

Table 4: molecular graphs

| Functional group | # Node | ACC |
|---|---|---|
| C-O(H) | 9 | 0.9207 |
| COO(H) | 121 | 0.9159 |
| $C_6(H_6)$ | 121 | 0.7245 |
| Mix | 121 | 0.8946 |

### 5.2    Attributed graphs

As shown in Theorem 3.5, the only condition required for Transformers to extract substructures is $\mathcal{T}(G, G') = 1$. Therefore, if node features are uniquely assigned to ensure a distinct graph representation for each graph, Transformers can extract substructures while incorporating these features, as discussed in Theorem D.16. Taking the AL-based feature description as an example, we define the attributed graph representation as $\mathsf{AL}_{\mathsf{f}}(G) = (v_1 f_1; ``:"; v_1^1 f_1^1; \cdots ; v_1^{m_1} f_1^{m_1}; ``,"; \cdots ; ``,"; v_n f_n; ``:"; v_n^1 f_n^1; \cdots ; v_n^{m_n} f_n^{m_n}).$ , where $f_i$ is the node features.

We use molecular graphs as examples to test how well Transformers understand attributed graphs with the attributed AL list $\mathsf{AL}_{\mathsf{f}}(G)$, where we set node feature $f_i$ as atoms. In the experiments, the transformers predict the positions of functional groups like Hydroxyl (C-O(H)), Carboxyl (COO(H)),

and Benzene Ring $C_6(H_6)$. We also use a mixed training setup where Hydroxyl and Carboxyl are combined as "Mix". Molecules contain 1 to 4 target groups. Details are in Appendix E.4.4 Results (see Table 4) show that Transformers perform well, even with mixed training, aligning with our discussion in Theorem 3.8 and Theorem 3.10.

## 6 Related work

**Evaluations for LLMs in graph understanding**   Benchmarks reveal that LLMs can recover graph structure from text. NLGraph showed basic reachability and shortest-path competence [21]; Instruct-Graph and GraphArena scaled tasks and graphs, with graph-aware verbalization and instruction-tuning boosting accuracy even on million-node inputs [22, 20]. GPT-4 few-shot can rival GNNs on node classification but is sensitive to token order [26]. Parallel work builds graph foundation models: GraphToken injects learned tokens, yielding substantial performance gains [32]; Graph2Token aligns molecules with text; and recent surveys chart cross-domain transfer [27, 23]. These two threads, task benchmarks and graph-biased LLMs, form the empirical backdrop for our ISF theory.

**Understanding transformers for graphs**   The mechanisms originate from graph learning methods, with detailed comparisons of graph neural networks, graph transformers, and decoder-only transformers in Appendix C. Theory now probes how vanilla Transformers perform graph reasoning. Log-depth models suffice, and are necessary, for connectivity and cycles [17], while width can trade for depth [29]. ALPINE [24] shows a GPT layer embeds adjacency and reachability, validating on planning tasks; two-layer decoders trained on shortest-path learn spectral line-graph embeddings instead of Dijkstra-style rules [5]. Surveys relate attention power to Weisfeiler–Lehman bounds and over-squashing limits [13, 19]; scalable variants such as AnchorGT mitigate $O(n^2)$ cost without losing accuracy [34]. Hierarchical distances or sparse global attention keep Transformers competitive on large or molecular graphs [9, 8, 30]. Collectively, these studies view attention heads as induced-neighbourhood selectors—exactly the mechanism ISF formalises via filtration depth.

## 7 Conclusion

This paper explores how decoder-only Transformers perform substructure reasoning over graphs represented as text. We propose ISF to model how substructures are progressively identified across layers. Our analysis shows that extraction accuracy depends on substructure size, model depth, and input format. We further validate ISF in LLMs, revealing consistent internal mechanisms. Extending this framework, we introduce the Tins method to handle composite and attributed graphs. These findings provide a unified view of how Transformers and LLMs reason over structured data.

## Acknowledgement

Xinnan Dai, Jay Revolinsky, Kai Guo, and Jiliang Tang are supported by the National Science Foundation (NSF) under grant numbers CNS2321416, IIS2212032, IIS2212144, IIS 2504089, DUE2234015, CNS2246050, DRL2405483 and IOS2035472, the Michigan Department of Agriculture and Rural Development, US Dept of Commerce, Gates Foundation, Amazon Faculty Award, Meta, NVIDIA, Microsoft and SNAP.

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

Table 5: Comparison among GNNs, Graph Transformers, and Decoder-Only Transformers (LLMs).

| | GNNs | Graph Transformer | Decoder-Only Transformer |
|---|---|---|---|
| **Input** | Discrete graph data (node features, adjacency matrix) | Discrete graph data (node features, adjacency matrix) | Textual description of a graph |
| **Output** | Scalar value (e.g., count) or fixed-size vector (e.g., for classification) | Scalar value (e.g., count) or fixed-size vector (e.g., for classification) | Text tokens forming a human-readable structural description in indefinite length |
| **Learning Formulation** | Encode graph via message passing and neighborhood aggregation; supervised on scalar outputs | Encode graph via graph-aware self-attention; supervised on scalar outputs | Next-token prediction over graph-structured text |
| **Mechanism** | 1-WL [4] | k-WL [14] / SEG-WL [33] | The proposed ISF (Ours) |

# Appendix

# A    Boarder Impact

In this work, we present new insights into how Transformers solve the substructure extraction task, offering a deeper understanding of their internal mechanisms. Our contributions span three key areas: (1) we introduce a novel concept of Induced Substructure Filtration (ISF), instructing LLMs in structure data understanding; (2) we propose a decomposition-based approach for tackling complex substructure reasoning by breaking down intricate patterns into simpler components, enabling more efficient extraction. This concept of thinking in substructures can generalize beyond graph tasks—for example, complementing step-by-step reasoning with pattern-by-pattern thinking; and (3) we provide both theoretical and empirical evidence supporting the development of graph foundation models, highlighting the potential of Transformers as backbones for structured learning tasks.

# B    Limitation

In this work, we focus primarily on the fundamentals of the substructure extraction task. However, other substructure-related tasks remain to be explored in future research. Additionally, our current study provides a high-level overview of decoder-only Transformers, leaving out theoretical details. Future work can extend this foundation to develop a more comprehensive and rigorous understanding.

# C    Related work

The mechanisms by which machine learning models learn graph-related problems have been widely studied, ranging from graph neural networks to transformers. However, due to their differing input–output formulations, the underlying mechanisms vary substantially, as summarized in Table 5.

# D    More on Theoretical Analysis

## D.1    Preliminaries

Let $G = (V, E)$ be a directed graph, where $V = \{v_1, \ldots, v_n\}$. For each vertex $v_i \in V$, denote $N(v_i) = \{v \in V \mid (v_i, v) \in E\} = \{v_i^1, \cdots, v_i^{m_i}\}$ as the set of its neighbors. We formally define two sequence representations of the graph $G$ where each vertex identifier $(v_i, v_i^j)$ and the special symbols (":"; ","; "|") are treated as individual tokens.

The adjacency list sequence representation $\mathsf{AL}(G)$ is constructed by concatenating blocks of tokens for each vertex $v_i$, separated by special token ",". The block for vertex $v_i$ consists of the token $v_i$ and token ":", followed by the sequence of tokens representing its neighbors $v_i^1, \ldots, v_i^{m_i}$. Formally, we have the following definition.

**Definition D.1** (Adjacency List Graph Representation). For a directed graph $G = (V, E)$ with $V = \{v_1, \cdots, v_n\}$, denote $N(v_i) = \{v \in V \mid (v_i, v) \in E\} = \{v_i^1, \cdots, v_i^{m_i}\}$ as the set of its

neighbors. The adjacency list graph representation of $G$ is defined as

$$\mathsf{AL}(G) = (v_1; \text{“:”}; v_1^1; \cdots; v_1^{m_1}; \text{“,”}; \cdots; \text{“:”}; v_n; \text{“:”}; v_n^1; \cdots; v_n^{m_n}).$$

The edge list sequence representation $\mathsf{EL}(G)$ is constructed by sequentially listing token pairs $(v_i, v_i^j)$ representing edges, separated by the "|" token. We give the formal definition below.

**Definition D.2** (Edge List Graph Representation). For a directed graph $G = (V, E)$ with $V = \{v_1, \cdots, v_n\}$, denote $N(v_i) = \{v \in V \mid (v_i, v) \in E\} = \{v_i^1, \cdots, v_i^{m_i}\}$ as the set of its neighbors. The edge list graph representation of $G$ is defined as

$$\mathsf{EL}(G) = (v_1; v_1^1; \text{“|”}; \cdots; \text{“|”}; v_1; v_1^{m_1}; \text{“|”}; \cdots; \text{“|”}; v_n; v_n^1; \text{“|”}; \cdots; \text{“|”}; v_n; v_n^{m_n}).$$

We also define adjacency matrix for graph $G$ as $A(G)$, where

$$A(G)_{i,j} = \begin{cases} 1, & (v_i, v_j) \in E \\ 0, & (v_i, v_j) \notin E \end{cases}.$$

In the subsequent analysis, we need a vectorized representation of a matrix or tensor. We first define tensor vectorization as follows.

**Definition D.3** (Tensor Vectorization). Let $\mathcal{A}$ be a $d$-dimensional tensor (or tensor of order $d$) with dimensions $(n_1, n_2, \ldots, n_d)$, denoted as $\mathcal{A} \in \mathbb{R}^{n_1 \times n_2 \times \cdots \times n_d}$. The elements of $\mathcal{A}$ are indexed by a tuple $(i_1, i_2, \ldots, i_d)$, where $1 \leq i_k \leq n_k$ for $k \in \{1, 2, \ldots, d\}$.

The vectorization of $\mathcal{A}$, denoted as $\mathrm{vec}(\mathcal{A})$, is a one-dimension vector $\mathbf{a} \in \mathbb{R}^N$, where $N = \prod_{k=1}^{d} n_k$ is the total number of elements in $\mathcal{A}$.

The elements of the vector $\mathbf{a} = (a_1, a_2, \ldots, a_N)$ are obtained by arranging the elements $\mathcal{A}_{i_1, i_2, \ldots, i_d}$ of the tensor $\mathcal{A}$ in row-major order. Specifically, the tensor element $\mathcal{A}_{i_1, i_2, \ldots, i_d}$ maps to the vector element $a_j$, where the index $j$ ($1 \leq j \leq N$) is determined by the following formula:

$$j = 1 + \sum_{m=1}^{d} \left( (i_m - 1) \prod_{l=m+1}^{d} n_l \right).$$

Here, the empty product convention is used, i.e., $\prod_{l=d+1}^{d} n_l \triangleq 1$.

*Remark* D.4. This indexing scheme corresponds to ordering the elements such that the last index $i_d$ varies the fastest, followed by the second-to-last index $i_{d-1}$, and so on, with the first index $i_1$ varying the slowest. For example, in the case of a matrix ($d = 2$), this corresponds to concatenating the rows of the matrix.

**Definition D.5** (Subgraph Isomorphism Indicator Tensor). Let $G = (V, E)$ and $G' = (V', E')$ be two directed graphs, referred to as the target graph and the query graph, respectively. Let $n = |V|$ and $k = |V'|$, and denote $V = \{v_1, v_2, \ldots, v_n\}$, $V' = (v_1', v_2', \ldots, v_k')$.

The subgraph isomorphism indicator tensor $\mathcal{T}(G, G')$ associated with $G, G'$, and the chosen vertex orderings is a $k$-dimensional tensor of size $n \times n \times \cdots \times n$. An element $\mathcal{T}(G, G')_{j_1, j_2, \ldots, j_k}$ of the tensor $\mathcal{T}(G, G')$, indexed by a tuple $(j_1, j_2, \ldots, j_k)$ where $1 \leq j_l \leq n$ for all $l \in \{1, \ldots, k\}$, satisfies:

$$\mathcal{T}(G, G')_{j_1, j_2, \ldots, j_k} \begin{cases} = 1, & \text{if the mapping } f : V' \to V \text{ defined by } f(v_l') = v_{j_l} \text{ for } l = 1, \ldots, k \\ & \quad \text{satisfies both conditions:} \\ & \quad \quad \text{(i) Injectivity: } v_{j_1}, v_{j_2}, \ldots, v_{j_k} \text{ are distinct vertices in } V \\ & \quad \quad \quad \text{(i.e., } j_l \neq j_m \text{ for all } 1 \leq l < m \leq k). \\ & \quad \quad \text{(ii) Edge Preservation: For every directed edge } (v_p', v_q') \in E', \\ & \quad \quad \quad \text{the directed edge } (f(v_p'), f(v_q')) = (v_{j_p}, v_{j_q}) \text{ exists in } E. \\ \leq 0, & \text{otherwise.} \end{cases}$$

That is, $\mathcal{T}(G, G')_{j_1, \ldots, j_k} = 1$ if and only if the sequence of target vertices $(v_{j_1}, \ldots, v_{j_k})$ forms a subgraph in $G$ that is isomorphic to $G'$ under the mapping implied by the indices and the fixed vertex orderings.

Throughout our theoretical analysis, we consider log-precision auto-regressive Transformer, instead of constant-precision Transformer. See [12, Appendix B] for more discussions.

### D.2 Technical Lemmas

**Lemma D.6** (Adjacency Matrix Extraction)**.**

>   (i) *For any integer $n$, there exists a two-layer log-precision Transformer with single attention head and hidden dimension $O(n^2)$, such that for any directed graph $G = (V, E)$ with $|V| = n$, the Transformer can output $\mathsf{vec}(A(G))$ for input sequence $\mathsf{AL}(G)$.*

>   (ii) *For any integer $n$, there exists a two-layer log-precision Transformer with single attention head and hidden dimension $O(n^2)$, such that for any directed graph $G = (V, E)$ with $|V| = n$, the Transformer can output $\mathsf{vec}(A(G))$ for input sequence $\mathsf{EL}(G)$.*

*Proof.* We need token embeddings to encode the index of the node ($i$ for vertex $v_i$), the type of the node ($v_i$ or $v_i^j$), and absolute positional embedding.

The first attention layer finds all the edges in $G$. For adjacency list graph representation, we COPY the value of $n \times (i-1)$ (from the position $v_i$) to the positions $v_i^1, \cdots, v_i^{m_i}$. Applying [11, Lemma C.7] and setting $\langle \boldsymbol{q}_i, \boldsymbol{k}_j \rangle = \|\boldsymbol{x}_i - \boldsymbol{x}_j\|_2^2$, where $\boldsymbol{x}_i$ is the type of the node in the embedding suffices. For edge list graph representation, it suffices to COPY from the previous token. Thus we can set $\langle \boldsymbol{q}_i, \boldsymbol{k}_j \rangle = (i - j - 1)^2$.

The subsequent MLP calculates $\mathsf{vec}(A(G[\{v_i, v_j\}]))$ for each edge. Notice that the value on index $k$ can be formulated as

$$\mathbf{1}_{k=n \times (i-1)+j} = \mathrm{ReLU}[k - n \times (i-1) - j + 1] + \mathrm{ReLU}[k - n \times (i-1) - j - 1] \\ - 2\mathrm{ReLU}[k - n \times (i-1) - j]. \tag{1}$$

By [11, Lemma C.2], Equation (1) can be calculated with constant hidden dimension.

The second attention layer aggregates all $\mathsf{vec}(A(G[\{v_i, v_j\}]))$ for each edge. We first calculate the MEAN of all valid $\mathsf{vec}(A(G[\{v_i, v_j\}]))$ from the last layer by [11, Lemma C.8]. The result can be expressed as $\mathsf{vec}(A'(G))$ where

$$A'(G)_{i,j} \begin{cases} \geq \frac{1}{n^2}, & (v_i, v_j) \in E \\ = 0, & (v_i, v_j) \notin E \end{cases}.$$

Thus we can get $A(G)_{i,j}$ by

$$A(G)_{i,j} = n^2 \cdot \mathrm{ReLU}\left( \frac{1}{n^2} - A'(G)_{i,j} \right),$$

which can be implemented in the subsequent MLP by [11, Lemma C.2]. $\qquad\square$

**Lemma D.7** (One-Step Subgraph Isomorphism Indicator Tensor Calculation)**.** *Fix integers $n, k, m \geq 1$. Let $V' = \{v_1', \ldots, v_k'\}$ be a set of $k$ vertices, and let $\mathcal{F}(V') = (V_1', V_2', \ldots, V_m')$ be a $k$-node $m$-filtration on $V'$ (defined in Definition 3.1). There exists $m$ two-layer MLPs with GeLU activation $\boldsymbol{f}_1, \cdots, \boldsymbol{f}_m$ each with hidden dimension $O(n^2 k^2)$, such that for any directed graphs $G = (V, E)$ with $V = \{v_1, \ldots, v_n\}$ and $G' = (V', E')$, $\boldsymbol{f}_i$ can output $\mathsf{vec}(\mathcal{T}(G, G'[V_i']))$ for input $\mathsf{vec}(A(G)) \oplus \mathsf{vec}(A(G')) \oplus \mathsf{vec}(\mathcal{T}(G, G'[V_{i-1}']))$ ($\oplus$ denotes concatenation), where $V_0' := \varnothing$.*

*Proof.* Denote $k_i = |V_i'|$. Without loss of generosity, we assume $V_i' = \{v_1, \cdots, v_{k_i}\}$. Notice that $\mathcal{T}(G, G_i')_{j_1, \cdots, j_{k_i}}$ can be calculated as

$$\mathcal{T}(G, G_{i-1}')_{j_1, \cdots, j_{k_{i-1}}} + \sum_{x,y \leq k_i;\ x > k_{i-1} \vee y > k_{i-1}} \left[ A(G')_{x,y} A(G)_{j_x, j_y} - A(G')_{x,y} \right] \\ - \mathbf{1}_{\exists 1 \leq x < y \leq k_i,\ j_x = j_y}.$$

Thus it suffices to calculate the value of $A(G')_{x,y} A(G)_{x',y'}$ for $1 \leq x, y \leq k$, $1 \leq x', y' \leq n$. By [11, Lemma C.1], this can be implemented with $O(n^2 k^2)$ hidden dimension. $\qquad\square$

### D.3 Proofs for Section 3

**Theorem 3.3** (Expressiveness for Progressive Identification)**.** *Given a $k$-node $m$-filtration $\mathcal{F}(V')$ on $V' = \{v_1', \ldots, v_k'\}$. For any directed graphs $G = (V, E)$ ($|V| = n$) and $G' = (V', E')$, a*

*log-precision Transformer with $m + 2$ layers, constant heads, and $O(n^k)$ hidden dimension can output* $\mathsf{vec}(\mathcal{T}(G, G'[V'_i]))$ *at layer* $i + 2$ *for* $i \in \{1, \ldots, m\}$.

**Theorem D.8** (Formal Statement of Theorem 3.3). *Fix integers $n \geq k \geq 1, m \geq 1$. Let $V' = \{v'_1, \ldots, v'_k\}$ be a set of $k$ vertices, and let $\mathcal{F}(V') = (V'_1, V'_2, \ldots, V'_m)$ be a $k$-node $m$-filtration on $V'$. There exists a log-precision Transformer with $m + 2$ layers, constant number of attention heads, and $O(n^k)$ hidden dimension, such that for any directed graphs $G = (V, E)$ with $V = \{v_1, \ldots, v_n\}$ and $G' = (V', E')$, the Transformer processing $G, G'$ can output $\mathsf{vec}(\mathcal{T}(G, G'[V'_i]))$ at layer $i + 2$ for $i \in \{1, \cdots, m\}$. Here, $\mathsf{vec}(\cdot)$ is the vectorization for a tensor (formal defined in Definition D.3).*

*Proof.* The first two layers of Transformer calculates $\mathsf{vec}(A(G))$ and $\mathsf{vec}(A(G'))$ for graph $G, G'$. The desired output is at the last token for each graph, respectively. By Lemma D.6, this can be implemented with $O(n^2)$ hidden dimension, regardless of the representation of $G, G'$.

For the next $m$ layers, we first apply [11, Lemma C.7] to COPY $\mathsf{vec}(A(G)), \mathsf{vec}(A(G'))$ in the attention layer. This can be implemented by adding special marks on the last token for each graph. In the subsequent MLP layer, we apply Lemma D.7 to calculate desired results with $O(n^2 k^2)$ hidden dimension. $\qquad\square$

**Theorem 3.5** (Expressiveness for Pattern Extraction). *Under Assumption 3.4, for directed graphs $G = (V, E)$ ($|V| = n$) and $G' = (V', E')$ ($|V'| = k$), a log-precision Transformer with constant depth, constant heads, and $O(n^k)$ hidden dimension can output the unique $k$-tuple of vertices $(v_{i_1}, \ldots, v_{i_k})$ for which $\mathcal{T}(G, G')_{i_1, \ldots, i_k} = 1$.*

**Theorem D.9** (Formal Statement of Theorem 3.5). *Fix integers $n \geq k \geq 1$, and let $V' = \{v'_1, \ldots, v'_k\}$. There exists a log-precision Transformer with constant depth, constant number of attention heads, and $O(n^k)$ hidden dimension, such that for any directed graphs $G = (V, E)$ with $V = \{v_1, \ldots, v_n\}$ and $G' = (V', E')$ satisfying Assumption 3.4, the Transformer processing $G, G'$ can output the unique tuple $(v_{i_1}, \ldots, v_{i_k})$ for which $\mathcal{T}(G, G')_{i_1, \cdots, i_k} = 1$.*

*Proof.* We first take $m = 1$ in Theorem 3.3, indicating that a constant depth Transformer can output $\mathsf{vec}(\mathcal{T}(G, G'))$ for any directed graphs $G, G'$. By Assumption 3.4, $\mathsf{ReLU}(\mathsf{vec}(\mathcal{T}(G, G')))$ is a one-hot vector and can be obtained with a two-layer MLP via [11, Lemma C.2].

Notice that

$$i_x = \sum_{1 \leq i_1, \cdots, i_k \leq n} i_x \cdot \mathsf{ReLU}(\mathsf{vec}(\mathcal{T}(G, G'))),$$

thus by linear transformation we can obtain $(i_1, \cdots, i_k)$ for which $\mathcal{T}(G, G')_{i_1, \cdots, i_k} = 1$, from the corresponding one-hot vector $\mathsf{ReLU}(\mathsf{vec}(\mathcal{T}(G, G')))$.

The final step is to output $(v_{i_1}, \ldots, v_{i_k})$ sequentially. This can be obtained by adding one-hot positional encoding in all the output tokens to determine the current output position. Therefore, the next token can be obtained by calculating the inner product between $(i_1, \cdots, i_k)$ and the positional encoding. Since all the steps can be finished in constant layers, we finished our proof. $\qquad\square$

**Theorem 3.8** (Expressiveness for Single-Shape-Multi-Num Extraction). *Fix integers $n \geq k \geq 1$. There exists a log-precision Transformer with constant depth, constant number of attention heads, and $O(n^k)$ hidden dimension that can complete Single-Shape-Multi-Num Extraction defined in Definition 3.7 for directed graphs $G = (V, E)$ ($|V| = n$) and $G' = (V', E')$ ($|V'| = k$).*

**Theorem D.10** (Formal Statement of Theorem 3.8). *Fix integers $n \geq k \geq 1$, and let $V' = \{v'_1, \ldots, v'_k\}$. There exists a log-precision Transformer with constant depth, constant number of attention heads, and $O(n^k)$ hidden dimension, such that for any directed graphs $G = (V, E)$ with $V = \{v_1, \ldots, v_n\}$ and $G' = (V', E')$, the Transformer processing $G, G'$ can output all the tuples $(v_{i_1}, \ldots, v_{i_k})$ for which $\mathcal{T}(G, G')_{i_1, \cdots, i_k} = 1$.*

*Proof.* Follow the proof of Theorem 3.5, we first calculate $\boldsymbol{v}^1 = \mathsf{ReLU}(\mathsf{vec}(\mathcal{T}(G, G')))$ which marks all feasible tuples with 1, and 0 otherwise.

Next, we calculate $\boldsymbol{v}^2$ where $\boldsymbol{v}_i^2 = \boldsymbol{v}_1^1 + \cdots + \boldsymbol{v}_i^1$ by linear projection, and define $\boldsymbol{v}_i^3 = \boldsymbol{v}_i^1 \boldsymbol{v}_i^2$. In $\boldsymbol{v}_i^3$, the feasible tuples are marked as $1, 2, \cdots$, and 0 otherwise. By [11, Lemma C.1], $\boldsymbol{v}^2, \boldsymbol{v}^3$ can be obtained with a two-layer MLP.

The final step is to determine which position in which tuple the next-token corresponds to. This can be obtained by adding special positional encoding in the outputs. When the Transformer need to output the $x$-th tuple, it can first obtain the corresponding one-hot vector by the following formula:

$$\text{ReLU}(\boldsymbol{v}^3 - x - 1) + \text{ReLU}(\boldsymbol{v}^3 - x + 1) - 2 \cdot \text{ReLU}(\boldsymbol{v}^3 - x),$$

then following with the proofs in Theorem 3.5 to output the current position. By [11, Lemma C.2], the above steps can be obtained with constant-layer MLPs, which concludes our proof. $\square$

**Theorem 3.10** (Expressiveness for Multi-Shape-Single-Num Extraction). *Fix integers $n \geq k \geq 1$. There exists a log-precision Transformer with constant depth, constant heads, and $O(n^k)$ hidden dimension that can complete Multi-Shape-Single-Num Extraction defined in Definition 3.9 for a directed graph $G = (V, E)$ ($|V| = n$) and any target subgraph $G' = (V', E')$ with $|V'| = k' \leq k$ satisfying Assumption 3.4.*

**Theorem D.11** (Formal Statement of Theorem 3.10). *Fix integers $n \geq k \geq 1$. There exists a log-precision Transformer with constant depth, constant number of attention heads, and $O(n^k)$ hidden dimension, such that for any directed graph $G = (V, E)$ (with $V = \{v_1, \ldots, v_n\}$) and $G' = (V', E')$ (with $V' = \{v'_1, \cdots, v'_{k'}\}$ where $k' \leq k$) satisfying Assumption 3.4, the Transformer processing $G, G'$ can output the unique tuples $(v_{i_1}, \ldots, v_{i_{k'}})$ for which $\mathcal{T}(G, G')_{i_1, \cdots, i_{k'}} = 1$.*

*Proof.* The proof is based on that of Theorem 3.5, and we extend $G'$ to $\hat{G}'$ with $k - k'$ extra isolated node.

Now, for more general case that $k' \leq k$, there may exist multiple tuples $(v_{i_1}, \cdots, v_{i_k})$ such that $\mathcal{T}(G, \hat{G}')_{i_1, \cdots, i_k} = 1$. However, by Assumption 3.4, all these tuples shares the same $i_1, \cdots, i_{k'}$. Therefore, we can first obtain a one-hot vector via

$$\text{ReLU}(\boldsymbol{v}^3) + \text{ReLU}(\boldsymbol{v}^3 - 2) - 2 \cdot \text{ReLU}(\boldsymbol{v}^3 - 1),$$

where $\boldsymbol{v}^3$ is defined in the proof of Theorem 3.8. Finally, it suffices to output the corresponding $(v_{i_1}, \cdots, v_{i'_k})$, which is similar to the final step of Theorem 3.5. $\square$

## D.4 Theoretical Results for Section 5.1

**Assumption D.12.** For directed graphs $G = (V, E)$ with $V = \{v_1, \cdots, v_n\}$, $G' = (V', E')$ with $V' = \{v'_1, \cdots, v'_k\}$ and $V'_1, \cdots, V'_t \subseteq V'$, and a collection of $t$ vertex subsets $V'_1, \ldots, V'_t \subseteq V'$. It is assumed that:

   (i) There exists a *unique $k$-tuple* of distinct vertex indices $(i_1, \ldots, i_k)$ from $\{1, \ldots, n\}$ such that $\mathcal{T}(G, G')_{i_1, \ldots, i_k} = 1$.

   (ii) For each $j \in \{1, \ldots, t\}$, there is a fixed constant $c \geq 1$ such that the number of distinct $|V'_j|$-tuples of distinct vertex indices $(i_1, \ldots, i_{|V'_j|})$ from $\{1, \ldots, n\}$ for which $\mathcal{T}(G, G'[V'_j])_{i_1, \ldots, i_{|V'_j|}} = 1$ is at most $c$.

**Theorem D.13.** *Fix integers $n \geq k \geq 1$ and $t \geq 1$. Let $G' = (V', E')$ be a fixed directed graph with $V' = \{v'_1, \ldots, v'_k\}$. Let $V'_1, \ldots, V'_t$ be a collection of subsets of $V'$ such that $G'$ is covered by the subgraphs induced by these subsets, meaning $V' = \bigcup_{j=1}^t V'_j$ and $E' \subseteq \bigcup_{j=1}^t E(G'[V'_j])$. Denote $q = \max_{j \in \{1, \ldots, t\}} |V'_j|$.*

*There exists a log-precision Transformer with constant depth, constant number of attention heads, and $O(n^q + c^t + c^2 t^2 n)$ hidden dimension, such that: For any directed graph $G = (V, E)$ (with $V = \{v_1, \ldots, v_n\}$) that, together with the predefined $G'$ and subsets $V'_1, \ldots, V'_t$, satisfies Assumption D.12 (where $c$ is the constant from Assumption D.12), the Transformer processing $G$ can*

   (i) *First, for each $j = 1, \ldots, t$, output a special token $\langle S_j \rangle$, then identify and output all distinct $|V'_j|$-tuples of vertices $(v_{i_1}, \ldots, v_{i_{|V'_j|}})$ from $G$ such that $\mathcal{T}(G, G'[V'_j])_{i_1, \ldots, i_{|V'_j|}} = 1$.*

   (ii) *Subsequently, output a special token $\langle \text{ANS} \rangle$ and the unique $k$-tuple of vertices $(v_{i_1}, \ldots, v_{i_k})$ from $G$ such that $\mathcal{T}(G, G')_{i_1, \ldots, i_k} = 1$.*

*Remark* D.14. If we assume $c, t$ are both constants, then the result becomes $O(n^q)$, which demonstrates the advantages of thinking in substructures.

*Remark* D.15. Theorem D.13 highlights a trade-off concerning the hidden dimension complexity, $O(n^q + c^t + c^2t^2n)$. This complexity is influenced by:

- $t$: the number of intermediate decomposition steps, or the CoT steps.

- $q$: the maximum size of any intermediate subgraph $G'[V_j']$ considered during these steps.

- $c$: the maximum number of instances (matches) in $G$ for any such intermediate subgraph $G'[V_j']$.

When $t$ increases (employing more, potentially smaller, intermediate steps), $q$ generally decreases. However, $c$ may increase, as simpler or smaller intermediate subgraphs could appear more frequently. In this scenario, the $n^q$ component of the hidden dimension tends to decrease, while the $c^t + c^2t^2n$ components are likely to increase.

Conversely, when $t$ decreases (employing fewer, potentially larger, intermediate steps), $q$ generally increases. Correspondingly, $c$ may decrease, as more complex or larger intermediate subgraphs could be less common. This tends to increase the $n^q$ component, while the $c^t + c^2t^2n$ components are likely to decrease.

Thus, the optimal decomposition strategy for minimizing the required hidden dimension depends on the interplay between these parameters, dictated by the specific problem structure.

*Proof.* We first design the necessary embeddings.

1. For the special token $\langle S_j \rangle$, we need a $q^2$ dimension vector representing $\mathsf{vec}(A(G[V_j']))$ (if $|V_j'| < q$, then we need to add $q - |V_j'|$ isolated nodes); and a $n^q$ dimension vector representing $\mathsf{vec}(\mathcal{T}^{(j)})$ where $\mathcal{T}^{(j)}$ is a $q$-dimensional tensor of size $n \times n \times \cdots \times n$ defined as:

$$\mathcal{T}^{(j)}_{i_1,\cdots,i_q} = \begin{cases} 0, & \forall 1 \leq x < y \leq |V_j'|, \ i_x \neq i_y; \ \forall |V_j'| < x \leq q, \ i_x = 1 \\ 1, & \text{otherwise} \end{cases}. \quad (2)$$

2. For the output answer tokens in step (i), we need a $ctn$ dimension vector. For $v_{i_x}$ in the $y$-th tuple for the subgraph induced by $V_j'$, the embedding satisfies: the value on the $cn(j-1)+x'$ dimension is $i_x$, while the others are 0. Here, $x'$ is the node $v_{i_x}$ corresponds to in origin $G'$ ($v_{x'}'$).

To get the desired output sequence, we need to complete the following tasks:

- **Task 1:** At the position of $\langle S_j \rangle$, we need to get all the tuples $(i_1, \ldots, i_{|V_j'|})$ for which $\mathcal{T}(G, G'[V_j'])_{i_1,\ldots,i_{|V_j'|}} = 1$.

- **Task 2:** At the position of $\langle \text{ANS} \rangle$, we need to get the *unique* tuple $(i_1, \ldots, i_k)$ for which $\mathcal{T}(G, G')_{i_1,\ldots,i_k} = 1$.

For task 1, the idea is similar to the proof of Theorem 3.8. We first use Lemma D.6 to extract $\mathsf{vec}(A(G))$ for input graph $G$, then apply [11, Lemma C.7] to COPY $\mathsf{vec}(A(G))$ to the current position. Next, we calculate $\mathsf{vec}(\mathcal{T}'(G, \hat{G}'[V_j']))$. Here, $\hat{G}'[V_j']$ is obtained by adding $q - |V_j'|$ isolated nodes on $G'[V_j']$; and

$$\mathcal{T}'\left(G, \hat{G}'[V_j']\right)_{i_1,\cdots,i_q} = \begin{cases} = 1, & \text{if } \mathcal{T}\left(G, G'[V_j']\right)_{i_1,\ldots,i_{|V_j'|}} = 1 \text{ and } i_{|V_j'|+1} = \cdots = i_q = 1 \\ \leq 0, & \text{otherwise} \end{cases}.$$

Thus, $\mathsf{vec}(\mathcal{T}'(G, \hat{G}'[V_j']))$ is a $n^q$ dimensional vector. Notice that $\mathcal{T}'(G, \hat{G}'[V_j'])_{i_1,\cdots,i_q}$ can be calculated as

$$\sum_{1 \leq x,y \leq q} \left[ A\left(\hat{G}'[V_j']\right)_{x,y} A(G)_{i_x,i_y} - A\left(\hat{G}'[V_j']\right)_{x,y} \right] - \mathcal{T}^{(j)}_{i_1,\cdots,i_q},$$

where $\mathcal{T}^{(j)}$ is defined in Equation (2). The following steps are similar to the proof of Theorem 3.8, while the only difference is we only want the first $|V'_j|$ dimension. This can be implemented by modifying the positional encoding to give the correct position for the next-token.

For task 2, we first aggregate all the previous $|V'_j|$-tuples for $j = 1, \cdots, t$ using MEAN operation in [11, Lemma C.8]. We then multiplies the result with the sequence length (which can be obtained by absolute positional encoding). After this, we get a $ctn$-dimension vector $(\boldsymbol{b}_{1,1}, \cdots, \boldsymbol{b}_{1,c}, \boldsymbol{b}_{2,1}, \cdots, \boldsymbol{b}_{t,1}, \cdots, \boldsymbol{b}_{t,c})$. Here, $\boldsymbol{b}_{i,j}$ is a $n$-dimension vector corresponds to the $j$-th tuple for the subgraph induced by $V'_i$.

Next, we maintain a $t$-dimension tensor $\mathcal{T}^{\text{ans}}$ of size $c \times c \times \cdots \times c$, defined as

$$
\mathcal{T}^{\text{ans}}_{i_1, \cdots, i_t} \begin{cases} = 0, & \text{if } \boldsymbol{b}_{1,i_1}, \cdots, \boldsymbol{b}_{t,i_t} \text{ can be combined as } G' \\ \geq 1, & \text{otherwise} \end{cases}.
$$

By Assumption D.12, there exists a *unique* $t$-tuple $(i_1, \ldots, i_t)$ such that $\mathcal{T}^{\text{ans}}_{i_1, \cdots, i_t} = 0$. Notice that $\boldsymbol{b}_{1,i_1}, \cdots, \boldsymbol{b}_{t,i_t}$ can be combined as $G'$ if and only if the following holds:

$$
\begin{cases} \forall 1 \leq x \leq t, \ \boldsymbol{b}_{x,i_x} \neq \boldsymbol{0} \\ \forall 1 \leq x < y \leq t, \ \forall 1 \leq z \leq n, \ (\boldsymbol{b}_{x,i_x})_z = (\boldsymbol{b}_{y,i_y})_z \text{ or } (\boldsymbol{b}_{x,i_x})_z = 0 \text{ or } (\boldsymbol{b}_{y,i_y})_z = 0 \end{cases} \quad (3)
$$

Since $(\boldsymbol{b}_{y,i_y})_z \in \{0, 1, \cdots, n\}$, Equation (3) is equivalent to

$$
\begin{cases} \forall 1 \leq x \leq t, \ \text{ReLU} \left[ 1 - \sum_{1 \leq z \leq n} (\boldsymbol{b}_{x,i_x})_z \right] = 0 \\ \forall 1 \leq x < y \leq t, \ \forall 1 \leq z \leq n, \ \text{ReLU}[(\boldsymbol{b}_{x,i_x})_z - (\boldsymbol{b}_{y,i_y})_z] + \text{ReLU}[(\boldsymbol{b}_{y,i_y})_z - (\boldsymbol{b}_{x,i_x})_z] = 0 \\ \qquad\qquad\qquad\qquad\quad \text{or } (\boldsymbol{b}_{x,i_x})_z = 0 \text{ or } (\boldsymbol{b}_{y,i_y})_z = 0 \end{cases}
$$

The second condition is equivalent to $\forall 1 \leq x < y \leq t, \ \forall 1 \leq z \leq n$,

$$
\text{ReLU} \left[ 1 - \text{ReLU}[(\boldsymbol{b}_{x,i_x})_z - (\boldsymbol{b}_{y,i_y})_z] - \text{ReLU}[(\boldsymbol{b}_{y,i_y})_z - (\boldsymbol{b}_{x,i_x})_z] \right]
$$
$$
+ \text{ReLU}[1 - (\boldsymbol{b}_{x,i_x})_z] + \text{ReLU}[1 - (\boldsymbol{b}_{y,i_y})_z] \geq 1,
$$

or

$$
\text{ReLU} \left[ 1 - \text{ReLU} \left[ 1 - \text{ReLU}[(\boldsymbol{b}_{x,i_x})_z - (\boldsymbol{b}_{y,i_y})_z] - \text{ReLU}[(\boldsymbol{b}_{y,i_y})_z - (\boldsymbol{b}_{x,i_x})_z] \right] \right.
$$
$$
\left. - \text{ReLU}[1 - (\boldsymbol{b}_{x,i_x})_z] - \text{ReLU}[1 - (\boldsymbol{b}_{y,i_y})_z] \right] = 0.
$$

Thus, for any tuple $(i_t, \cdots, i_t)$, we can get $\mathcal{T}^{\text{ans}}_{i_1, \cdots, i_t}$ via an MLP with constant layers and $O(nt^2)$ hidden dimension. Notice that there are many components remaining the same when calculating different $\mathcal{T}^{\text{ans}}_{i_1, \cdots, i_t}$. We can calculate

$$
\text{ReLU} \left[ 1 - \text{ReLU} \left[ 1 - \text{ReLU}[(\boldsymbol{b}_{p_1,q_1})_z - (\boldsymbol{b}_{p_2,q_2})_z] - \text{ReLU}[(\boldsymbol{b}_{p_2,q_2})_z - (\boldsymbol{b}_{p_1,q_1})_z] \right] \right.
$$
$$
\left. - \text{ReLU}[1 - (\boldsymbol{b}_{p_1,q_1})_z] - \text{ReLU}[1 - (\boldsymbol{b}_{p_2,q_2})_z] \right]
$$

for all $(p_1, q_1), (p_2, q_2)$ pairs and $1 \leq z \leq n$, which are $O(c^2 t^2 n)$. Each can be calculated via an MLP with constant depth and constant hidden dimension.

Finally, we will calculate the *unique* $t$-tuple $(i_1, \ldots, i_t)$ such that $\mathcal{T}^{\text{ans}}_{i_1, \cdots, i_t} = 0$. Notice that

$$
i_x = \sum_{1 \leq i_1, \cdots, i_t \leq c} \text{ReLU}(1 - \mathcal{T}^{\text{ans}}_{i_1, \cdots, i_t}) \cdot \left( \sum_{1 \leq j \leq t} \boldsymbol{b}_{j,i_j} \right),
$$

which can be calculated via an MLP with constant depth and $O(c^t)$ hidden dimension by [11, Lemma C.1]. $\qquad\square$

## D.5 Theoretical Results for Section 5.2

**Theorem D.16.** *Fix integers $n \geq k \geq 1$, and let $V = \{v_1, \cdots, v_n\}, V' = \{v'_1, \cdots, v'_k\}$. Fix a feature function $\varphi : V \cup V' \to \mathbb{Z}$. There exists a log-precision Transformer with constant depth, constant number of attention heads, and $O(n^k)$ hidden dimension, such that for any directed*

| heads | embedding | drop out rate | batch size | learning rate | max epoch |
|---|---|---|---|---|---|
| 12 | 384 | 0.2 | 2048 | 0.001 | 40000 |

graphs $G = (V, E)$ and $G' = (V', E')$, the Transformer processing $G, G'$ can output all the tuples $(v_{i_1}, \cdots, v_{i_k})$ that satisfy both of the following conditions:

(i) *Subgraph Isomorphism: The subgraph of $G$ induced by the set of vertices $\{v_{i_1}, \cdots, v_{i_k}\}$ is isomorphic to $G'$ under the mapping $v'_p \mapsto v_{i_p}$ for $p \in \{1, \cdots, k\}$. That is, $\mathcal{T}(G, G')_{i_1, \cdots, i_k} = 1$.*

(ii) *Feature Matching: For all $p \in \{1, \ldots, k\}$, the feature of the $p$-th vertex in the tuple from $G$ matches the feature of the $p$-th vertex in $V'$, i.e., $\varphi(v_{i_p}) = \varphi(v'_p)$.*

*Proof.* The proof is similar to that of Theorem 3.8. We define a $k$-dimensional tensor of size $n \times n \times \cdots \times n$ $\mathcal{T}'(G, G')$ as

$$
\mathcal{T}'(G, G')_{j_1, \cdots, j_k}
\begin{cases}
= 1, & \text{if the mapping } f : V' \to V \text{ defined by } f(v'_l) = v_{j_l} \text{ for } l = 1, \ldots, k \\
& \text{satisfies both conditions:} \\
& \quad \text{(i) Injectivity: } v_{j_1}, v_{j_2}, \ldots, v_{j_k} \text{ are distinct vertices in } V \\
& \quad \text{(i.e., } j_l \neq j_m \text{ for all } 1 \leq l < m \leq k\text{).} \\
& \quad \text{(ii) Edge Preservation: For every directed edge } (v'_p, v'_q) \in E', \\
& \quad \text{the directed edge } (f(v'_p), f(v'_q)) = (v_{j_p}, v_{j_q}) \text{ exists in } E. \\
& \quad \text{(iii) Feature Matching: For all } p \in \{1, \cdots, k\}, \text{ the features of the} \\
& \quad p\text{-th vertex match, i.e., } \varphi(v_{i_p}) = \varphi(v'_p). \\
\leq 0, & \text{otherwise.}
\end{cases}
$$

Notice that $\mathcal{T}'(G, G')_{j_1, \cdots, j_k}$ can be obtained as

$$
\mathcal{T}(G, G')_{j_1, \cdots, j_k} - \sum_{1 \leq x \leq k} \mathbf{1}_{\varphi(v_{i_x}) \neq \varphi(v'_x)},
$$

or

$$
\mathcal{T}(G, G')_{j_1, \cdots, j_k} - \sum_{1 \leq x \leq k} \left[ \text{ReLU}(\varphi(v_{i_x}) - \varphi(v'_x)) + \text{ReLU}(\varphi(v'_x) - \varphi(v_{i_x})) \right].
$$

Therefore, it suffices to COPY the feature while constructing the adjacency matrix $A(G), A(G')$. And it suffices to further calculate the value of ReLU $\left[ \varphi(v_i) - \varphi(v'_j) \right]$, ReLU $\left[ \varphi(v'_j) - \varphi(v_i) \right]$, which requires $O(nk)$ hidden dimension in total (by [11, Lemma C.2]). $\qquad \square$

# E    Experiments setting

Here, we provide the details of our experimental setup. We use a lightweight version of the GPT-2 model, which is an implementation version of nano-GPT, with hyperparameters listed in Table 6. 10% of the data is used for validation, and the model is saved when the validation loss reaches its minimum. All experiments are conducted on a machine equipped with 8 NVIDIA A6000 GPUs.e.

## E.1    training details in input formulations

We take more 50,000 graphs for training and testing. Each graph contains a target substructure: either a triangle, square, or pentagon. While the number of training samples varies, the test set size remains fixed, as shown in Table 7. Since this is a toy example, we set the Transformer's hidden dimension to a small size of 192.

Table 7: The dataset information for the AL and EL comparison

|  | #Training data | #Test data | #Node |
|---|---|---|---|
| Triangle | 5000 | 1000 | 5 |
| Square | 15000 | 1000 | 8 |
| Pentagon | 35000 | 1000 | 8 |

Table 8: Performance across epochs for Square (4 layers) and Pentagon (5 layers).

| Epoch | 10000 | 20000 | 30000 | 40000 | 50000 | 60000 |
|---|---|---|---|---|---|---|
| **Square (4 layers)** | | | | | | |
| AL | $0.97 \pm 0.004$ | $0.98 \pm 0.003$ | – | – | – | – |
| EL | $0.83 \pm 0.078$ | $0.93 \pm 0.050$ | $0.99 \pm 0.006$ | – | – | – |
| **Pentagon (5 layers)** | | | | | | |
| AL | $0.69 \pm 0.017$ | $0.73 \pm 0.058$ | $0.84 \pm 0.031$ | $0.92 \pm 0.004$ | – | – |
| EL | $0.61 \pm 0.017$ | $0.60 \pm 0.019$ | $0.74 \pm 0.018$ | $0.87 \pm 0.010$ | $0.89 \pm 0.0056$ | $0.93 \pm 0.044$ |

## E.2 Multi-Shape setting

To evaluate the discrimination ability of Transformers in detecting multiple structures, we set the evaluations from four perspectives: 1. different numbers of nodes (Triangle vs. Square); 2. the same number of nodes but different numbers of edges (Square vs. Diamond); 3. the same number of nodes and edges, but different edge directions (F-triangle vs. T-triangle); 4. whether the substructure forms a closed loop (Square vs. Path). We construct 600K question-answer pairs to train a 4-layer transformer model. Since triangles require less training data, as suggested in the Multi-num task, we set the training sample ratio of Triangle to Square to 1:6, while maintaining a 1:1 ratio for the other substructure pairs.

## E.3 Efficient of EL and AL

EL performs worse than AL when trained for the same number of epochs, but it eventually reaches comparable performance. In our results, we selected the epoch at which AL achieves its best performance. However, we will also provide additional information indicating when EL catches up with AL, as shown in the Table 8 below:

EL with longer input lengths requires more training epochs to achieve the same performance as AL. Although EL and AL are theoretically equivalent in their ability to represent graph structures, the longer input sequences in EL lead to less efficient learning. We will clarify this point in the revision.

## E.4 LLMs Experiments

### E.4.1 Evaluation on substructure detection

In substructure detection, we set the question prompt as:

*Given a structure G, Node 1 is connected to Node 2, 3; Node 2 is connected to.... List all of the square patterns in the graph in the form of: [#1, #2, #3, ...]*

Meanwhile, we set the answer as:

*The answer is [1, 2, 3]*

For the triangle detection task, we use 1,000 training samples and 200 for evaluation. Using supervised fine-tuning (SFT) over 4 epochs, we achieve 58.86% accuracy on the test set. The model responses suggest that LLaMA3.1-8B-Instruct still generates explanatory content, including code, during answer generation.

We also evaluate LLM performance on the square detection task using 283 test samples, which contain only four distinct answer types. As shown in the Table 9, lightweight LLMs fail to extract meaningful patterns without fine-tuning. Due to the high computational cost of training LLMs, we

Table 9: Large Language Models do the ISF process in the middle layers

| Model | Llama3.2-3B-Instruct | Qwen3-4B-Base | Llama3.1-8B-Instruct |
|---|---|---|---|
| ACC / finetuned ACC | 0.0035 / 0.4982 | 0.0141 / 0.5724 | 0.0035/0.6572 |
| Vis. for non-finetuned | | | |
| Vis. for finetuned | | | |

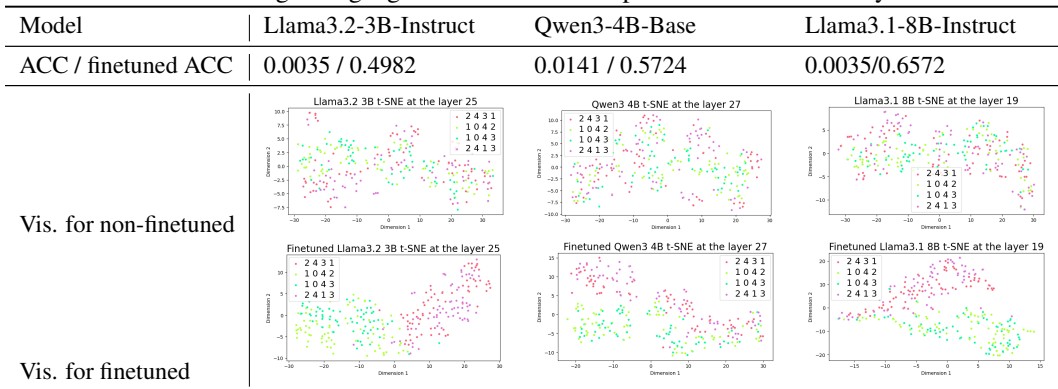

Table 10: Concept learning

| | no-finetune | without topos | with topos |
|---|---|---|---|
| llama | 0 | 0 | 0.50 |
| Qwen1.5 | 0 | 0 | 0.75 |
| Qwen3B | 0 | 0 | 0.85 |

limit the training data to only include samples with these four answer types, using 480 samples for training and 120 for validation. After fine-tuning, we observe a significant improvement in accuracy. Additionally, visualization shows that graphs corresponding to similar answers tend to cluster together.

### E.4.2 Evaluation on question prompt

e evaluate how LLMs align conceptual descriptions with underlying topological structures. Specifically, we test LLaMA3.2-3B-Instruct, Qwen2.5-1.5B, and Qwen2.5-3B by setting the temperature to 0.6, running 20 dialogue turns per model, and manually evaluating the responses.

First, we assess whether the models understand the concept of a "house" in graph terminology by prompting them with: *What is house in graphs? Giving the graph in the formulate of: 'Node 1 is connected to nodes 2, 3'* In the baseline setting (without topological prompts), we explicitly teach the concept using natural language definitions generated by Gemini-1.5-Pro. We fine-tune each model using 200 such concept-descriptive sentences. For example, a training sample might look like: *In graph theory, a "house" isn't a standard term like "tree" or "cycle." It usually refers to a specific small graph resembling a house drawing. This graph consists of five vertices and six edges. It's formed by a cycle of four vertices (the "walls" and "floor") with an additional vertex connected to one of the cycle vertices (the "roof peak").* In the train with topos setting, we add a topology description to the house, which is: *The house is described as: G describes an undirected graph among 1, 2, 3, 4, and 5.In this graph: Node 1 is connected to nodes 2, 5.Node 2 is connected to nodes 1, 3, 5. Node 3 is connected to nodes 2, 4. Node 4 is connected to nodes 3, 5. Node 5 is connected to nodes 1, 2, 4.*

As shown in Table 10, the LLMs only learn the topological descriptions training with the terminology terms together. The LLMs do not generate new concepts by the already known knowledge.

### E.4.3 Thinking-in-substructures

We use 100K samples for training and 5,000 for testing. Since each composite substructure is composed of different sets of decomposing substructures, the required thinking length varies accordingly. A summary of these decompositions is provided in Table 11

Table 11: Max length for each composite substructure extraction

| Substructures | $|\{P_1\}|$ | $|\{P_2\}|$ | $|\{P_3\}|$ | overall length |
|---|---|---|---|---|
| Diagnoal | 95 | 55 | - | 150 |
| Diamond | 55 | 95 | - | 150 |
| House | 75 | 115 | - | 190 |
| Complex | 80 | 100 | 110 | 290 |

### E.4.4 Transformers for moleculars

In this subsection, we introduce the experimental setup for applying transformers to molecular data. Specifically, we focus on the task of functional group recognition, where the goal is to identify the atomic positions corresponding to specific functional groups within molecules. We then introduce the dataset, functional group and experimental dataset construction.

**Dataset** We conduct experiments on QM9 [16] and PCBA [28]. The QM9 dataset primarily contains quantum mechanical calculated properties of approximately 134,000 molecules, suitable for molecular property prediction and quantum chemistry research. The PCBA dataset, on the other hand, contains activity data for approximately 440,000 molecules against 128 biological assays, making it more suitable for drug screening and bioactivity prediction.

**Functional Group** We search for molecules containing basic functional groups in the QM9 and PCBA datasets. Specifically, we extract 33,000 molecules containing hydroxyl groups (C-O-H) from QM9, and 13,000 molecules containing carboxyl groups (-COOH) as well as 33,000 molecules containing benzene rings ($C_6H_6$) from the PCBA dataset. For all these molecules, H atoms are ignored during processing. The maximum number of atoms is 9 for molecules containing hydroxyl groups, while it is 121 for both carboxyl- and benzene-containing molecules.

**Experimental Dataset Construction** For the hydroxyl group identification task, we first convert molecular graphs into molecular description inputs by omitting H atom. A simple example of such a description is "0 C : 1 O", and the corresponding answer for the position of the C–O–(H) group is "0,1". We then select molecules containing hydroxyl groups, using 30,000 molecules for the training set and 3,000 for the test set.

Similarly, for the identification of molecules containing carboxyl groups and benzene rings, we also convert molecular graphs into molecular description inputs by omitting hydrogen atoms, and generate the corresponding position answers for the target functional groups. We use 10,000 molecules for training and 3,000 for testing in the carboxyl group recognition task. For the benzene ring recognition task, we construct a dataset with 30,000 molecules for training and 3,000 for testing. The maximum input lengths for molecular descriptions are 100, 1000, and 1000 for molecules containing hydroxyl group, carboxyl group, and benzene ring, respectively.

In addition, we construct a mixed dataset containing molecules with hydroxyl and carboxyl groups. Specifically, we use 10,000 hydroxyl-containing molecules and 10,000 carboxyl-containing molecules for training, and 1,500 molecules of each type for testing.

