# OpenReview forum: "From Sequence to Structure: Uncovering Substructure Reasoning in Transformers"
_NeurIPS.cc/2025/Conference — NeurIPS 2025 poster_

### Official Review · Reviewer_J4XB · 2025-07-01

**Clarity:** 3
**Significance:** 2
**Originality:** 2
**Rating:** 4
**Confidence:** 4

**Summary:**

This work explores how substructure reasoning is enabled in Transformers when graph-structured inputs are presented as sequences, which has been a common approach for leveraging recent LLMs for graphs (text-attributed graphs mainly, but this paper presents a general case). It introduces Induced Substructure Filtration (ISF), a theoretical framework explaining how Transformers progressively identify substructures through layer-wise node aggregation. They provide formal theorems, empirical validation on both small trained models and LLMs (Llama), and extend the approach to complex multiple patterns and molecular graphs using a 'thinking in substructures' method. Overall, the paper aims to uncover some of the capabilities of LLMs to understand graph structures when the corresponding input is sequential and the presented insights seem useful, although understood in literature through the fact that (i) Transformers can be instantiated as a case of GNNs, and (ii) encode-process-decode networks are used in graph reasoning domains (e.g., neural algorithmic reasoning) to model progressive node aggregation.

**Questions:**

How does a Transformer trained with a substructure counting-heavy approach/dataset affect downstream performance in real-world tasks that take textified graphs as input?

**Ethical Concerns:**

["NO or VERY MINOR ethics concerns only"]

**Final Justification:**

The rebuttal demonstrated how their approach contributes to addressing the main question posed in the paper, a point I initially had concerns about due to closer prior works which investigates links between GNNs/Transformers, particularly regarding how such methods, despite not explicitly using graph structure, can still capture and understand the underlying graph.

**Quality:**

3

**Strengths And Weaknesses:**

Strengths:
- The proposed ISF gives an understanding of how Transformers or sequence models process graph structure.
- Empirical results are interesting and shows the challenges of large scale multi-resolution understanding in graph structures.
- The paper overall has neat visualization and interpretability of the messages.
- Insights such as the use of AL vs. EL in the text,graph prompt design is a useful message and is clearly demonstrated both conceptually and empirically.

Weaknesses:
- I believe one of the central messages of the paper on progressive node aggregation in Transformers is well known. Transformers are considered special cases of GNNs, which implies the understanding that a node must have at least m GNN operation layers to incorporate information from an m-hop receptive field. This is close to the result that Transformers progressively identify substructures through layer-wise node aggregation. In addition, the encode-process-decode framework in [1] suggests the GNN core be iterated for M times to have the GNN form progressive embeddings required for graph reasoning. This is used in neural algorithmic reasoning works, e.g., [2]. Therefore, the overall advances from this paper, apart from the methods and results in sections 5.1 and 5.2, may be limited, since the question of how ISF would apply to other structured reasoning tasks beyond substructure extraction (such as problem solving in maths, logic, etc.) remains open.
- minor typo on line 81 in page 3: there may be an 'convert' or similar word missing here "To input graphs into a Transformers, ..."

[1] Battaglia, P.W., Hamrick, J.B., Bapst, V., Sanchez-Gonzalez, A., Zambaldi, V., Malinowski, M., Tacchetti, A., Raposo, D., Santoro, A., Faulkner, R. and Gulcehre, C., 2018. Relational inductive biases, deep learning, and graph networks.
[2] Veličković, P. and Blundell, C., 2021. Neural algorithmic reasoning.

---

> ### Author Rebuttal · Authors · 2025-07-30
>
> Overall: Thanks for your suggestions. We would like to clarify your key concerns below.
>
> We respectively point out potential misunderstandings in your comments regarding the motivation and the goal of our paper. We want to clarify them below:
>
> Recent studies have shown that LLMs possess strong capabilities in graph reasoning tasks [3, 4, 5]. At the same time, it is important to understand the underlying mechanisms behind this graph reasoning ability with the textual data [6, 7]. We focus on the substructure extraction task, motivated by prior work [4, 5] showing that LLMs can identify and extract specific substructures from a given graph.
>
> To bridge the gap, the goal of our paper is to provide **insights into why LLMs can extract substructures from text descriptions of graphs**. As clearly stated in the abstract **How can a decoder-only Transformer architecture understand underlying graph structures?**. We do not aim to argue using transformers to replace GNNs or graph transformers, nor do we propose new algorithms to solve graph tasks.
>
> To this end, we train **decoder-only** transformers using a **text-based input** format, following the GPT-2 training pipeline, to explore the mechanisms underlying their graph reasoning abilities. Therefore, connecting and comparing with graph-specific encode models such as GNNs and graph transformers are completely beyond the scope of this work.
>
> - W: Transformers are considered special cases of GNNs, and the findings are not new.
>
> A: As discussed in the “Overall Response,” our work focuses on understanding why **decoder-only** transformers can capture substructures when trained using next-token prediction. Therefore, a detailed comparison with encoder-based frameworks falls outside the scope of this paper.
>
> Although it is off topic of our paper, we'd like to introduce the differences between decoder-only transformer and encoder based GNNs in the Table:
>
> |                  | GNNs                                                                                          | Decoder-Only Transformers (LLMs)                            |
> |--------------------------|-----------------------------------------------------------------------------------------------|-------------------------------------------------------------|
> | Input                    | Discrete graph data (node features, adjacency matrix)                                         | Textual description of a graph                              |
> | Output                   | A scalar value (e.g., count) or fixed-size vector (e.g., for classification)                  | Text tokens forming a human-readable structural description |
> | Learning Formulation     | Encode graph via message passing and neighborhood aggregation, then predict scalar outputs     | Next-token prediction over graph-structured text            |
> | Substructure Mechanism   | 1-WL [8]                                                                                         | ISF                                                         |
> | Information Aggregation  | m-hop                                                                                         | m-filter                                                    |
>
> The concept of information aggregation in our work, referred to as m-filter, is fundamentally different from the conventional m-hop aggregation in GNNs.
> While GNNs aggregate information from m-hop neighbors based on the adjacency matrix toward a central node, decoder-only transformers instead learn to filter relevant information across m tokens in the input sequence, without relying on explicit graph structure.
>
> - W: Comapring with the GNNs that used in reasoning tasks.
>
> A: As mentioned in the "Overall Response," this paper focuses on the question: "How can a decoder-only transformer understand graph structure?" Accordingly, our analysis is centered specifically on decoder-only transformers, which align with the architecture of LLMs. To the best of our knowledge, this is the first work to analyze structure understanding in the context of next-token prediction.
>
> [1] proposes graph neural network blocks to encode algorithmic reasoning tasks using edge and node representations in the form of vector lists, but our approach relies solely on textual sequences and trained by next token prediction. [2] presents a blueprint for neural algorithmic reasoning within an encoder-decoder framework, whereas our study targets decoder-only transformers. Therefore, both works are outside the scope of our discussion, which explains how decoder-only transformers trained on textual graph data can learn to understand structural information.
>
> In Sections 5.1 and 5.2, we continue to follow our central focus on how LLMs can understand graph structures presented in textual descriptions. These sections include more complex graph types, such as composed and attributed graphs. Our motivation for analyzing these graphs stems from prior findings that LLMs are capable of decomposing graphs and understanding attributed graphs such as molecular structures [5]. Thus, exploring neural algorithmic reasoning in real-world applications is beyond the scope of our current discussion.
>
> However, we would like to share our view that reasoning tasks often involve substructure understanding, making this an important direction for future research. Given that decoder-only transformers such as GPT-2 have demonstrated strong performance on reasoning tasks (e.g., mathematics and logic), it would be valuable to investigate whether graph reasoning can be framed as a specific instance of general reasoning in LLMs.
>
> - W: minor typo
>
> A: Thanks for your advice, we will change it into "Following prior work, we represent graph data using either Adjacency Lists (AL) or Edge Lists (EL), and encode them as textual sequences for input into Transformers" to make it clear.
>
> - Q: transformer trained with a substructure counting-heavy approach.
>
> A: We do not train the transformer to count substructures. Instead, we train a decoder-only transformer using a next-token prediction objective to extract substructure node lists from given graphs. While prior work using GNNs or graph transformers has primarily focused on substructure counting [8], our model is trained to generate complete substructure lists in text form. Previous studies have shown that the graph reasoning ability of decoder-only LLMs can be enhanced through cross-task fine-tuning on graph-related tasks [9, 10]. Building on this, our work provides theoretical foundations for why LLMs can understand structural information, offering a basis for extending their capabilities to a wide range of structure-based tasks.
>
> [1] Battaglia, P.W., Hamrick, J.B., Bapst, V., Sanchez-Gonzalez, A., Zambaldi, V., Malinowski, M., Tacchetti, A., Raposo, D., Santoro, A., Faulkner, R. and Gulcehre, C., 2018. Relational inductive biases, deep learning, and graph networks.
>
> [2] Veličković, P. and Blundell, C., 2021. Neural algorithmic reasoning.
>
> [3] Wang H, Feng S, He T, et al. Can language models solve graph problems in natural language?[J]. Advances in Neural Information Processing Systems, 2023, 36: 30840-30861.
>
> [4] Tang J, Zhang Q, Li Y, et al. GraphArena: Evaluating and Exploring Large Language Models on Graph Computation[C]//The Thirteenth International Conference on Learning Representations.
>
> [5] Dai X, Qu H, Shen Y, et al. How Do Large Language Models Understand Graph Patterns? A Benchmark for Graph Pattern Comprehension[C]//The Thirteenth International Conference on Learning Representations.
>
> [6] Saparov A, Pawar S A, Pimpalgaonkar S, et al. Transformers Struggle to Learn to Search[C]//The Thirteenth International Conference on Learning Representations.
>
> [7] Wang S, Shen Y, Feng S, et al. Alpine: Unveiling the planning capability of autoregressive learning in language models[J]. Advances in neural information processing systems, 2024, 37: 119662-119688.
>
> [8] Chen Z, Chen L, Villar S, et al. Can graph neural networks count substructures?[J]. Advances in neural information processing systems, 2020, 33: 10383-10395.
>
> [9] Zhang Q, Chen N, Li Z, et al. Improving LLMs' Generalized Reasoning Abilities by Graph Problems[J]. arXiv preprint arXiv:2507.17168, 2025.
>
> [10] Zhang Y, Wang H, Feng S, et al. Can LLM Graph Reasoning Generalize beyond Pattern Memorization?[C]//Findings of the Association for Computational Linguistics: EMNLP 2024. 2024: 2289-2305.

---

> > ### Author Response · Authors · 2025-08-06
> >
> > We deeply appreciate your effort. As we approach the end of the discussion period, we summarize our key clarifications below for your convenience:
> >
> > - Motivation and Scope – Our goal is to understand why decoder‑only Transformers can extract substructures from textual graph descriptions. We do not aim to replace or compare with GNNs/graph transformers, nor do we propose new algorithms for solving graph tasks. We clarified this explicitly in the “Overall Response” and added a comparison table highlighting the fundamental differences between decoder‑only Transformers and GNNs.
> >
> > - Relation to GNNs in Reasoning Tasks – While prior works focus on encoder‑based frameworks for neural algorithmic reasoning, our analysis is centered on decoder‑only Transformers trained with next‑token prediction. This is, to our knowledge, the first study to analyze structure understanding in this specific setting.
> >
> > - Generality of ISF and Tins – We use simplified textual graph datasets, following prior work, to capture key properties of decoder‑only Transformers.
> >
> > - Counting‑Heavy Approach – Our models are not trained for substructure counting; instead, they generate complete node lists for each substructure in text form, a formulation that captures sequential and structural dependencies and differs fundamentally from prior counting‑based GNN methods.
> >
> >
> > We would greatly appreciate it if you could review these clarifications and let us know whether they address your concerns. Any additional questions or feedback during the remaining discussion period would be most welcome, as they will help us further strengthen the final version.

---

> ### Author Response · Authors · 2025-08-07
>
> Dear reviewer, thank you for your feedback. We hope we've addressed all of your concerns. As we approach the end of the discussion period, we still welcome further discussions. Please feel free to let us know if you have any additional questions or concerns.

---

> > ### Comment · Reviewer_J4XB · 2025-08-08
> >
> > Thank you for the detailed response and clarifications to my comments. While I appreciate the distinctions you raised regarding the related literature and the core scientific understanding of the main question posed in the paper, as well as your points on the scope, focus, and methodology adopted in this work, I believe the main question “How can a decoder-only Transformer architecture understand underlying graph structures?” can be understood considering the scientific understanding presented in the prior works I mentioned. However, based on the merits of the methodology as well as the important distinction regarding the 'consideration of graph adjacency matrix' in prior references and not this work, I improve my score. I also hope the manuscript can be revised to clarify such distinctions in appropriate places of positioning.

---

> > > ### Author Response · Authors · 2025-08-08
> > >
> > > Thank you for your response and valuable suggestions. We are pleased to have addressed all of your concerns. We will revise the paper to enhance its clarity and more clearly differentiate it from the prior work you mentioned.

---

### Official Review · Reviewer_ZFQo · 2025-07-01

**Clarity:** 3
**Significance:** 3
**Originality:** 3
**Rating:** 4
**Confidence:** 3

**Summary:**

This paper explores the mechanisms by which decoder-only Transformers, including large language models (LLMs), perform substructure extraction and reasoning on graph-structured data represented as text. The authors introduce the novel concept of Induced Substructure Filtration (ISF) to interpret how Transformers progressively identify graph substructures. The work includes both theoretical modeling and empirical studies and is extended to consider composite and attributed graphs via the Thinking-in-Substructure (Tins) approach.

**Questions:**

1.	How does the proposed ISF framework compare empirically with existing graph reasoning models, such as GNNs or Transformer variants like Graphormer, on standard graph benchmarks?
2.	To what extent are the results sensitive to the format and structure of the input prompts?
3.	How scalable is the proposed method (ISF and Tins) when applied to large graphs with high substructure complexity in real-world settings?

**Ethical Concerns:**

["NO or VERY MINOR ethics concerns only"]

**Limitations:**

Yes.

**Quality:**

3

**Strengths And Weaknesses:**

**Strength**

1.	This paper proposes the concept of ISF, providing a mathematically grounded explanation for how Transformers identify sub-structures layer by layer, with a solid theoretical foundation.
2.	Extensive experiments conducted on a variety of synthetic datasets demonstrate the effectiveness of the method.
3.	The paper delves into the relationship between Transformers trained from scratch and LLMs fine-tuned with instructions.

**Weakness**

1.	The paper does not include comparisons with existing GNNs or graph-aware Transformer architectures on the substructure extraction task.
2.	The experiments mainly focus on synthetic datasets and small molecular graphs. It is unclear how well the proposed methods generalize to real-world graphs with larger size and noise.
3.	The applicability of the ISF framework to Transformer variants with different architectures or to other domains is not fully discussed.

---

> ### Author Rebuttal · Authors · 2025-07-30
>
> Overall response: Thank you for your suggestions. We would like to clarify your key concerns below
>
> - W: decoder-only transformers Vs. GNNS and graph transformers
>
> A: We respectively point out potential misunderstandings in your comments regarding the motivation and the goal of our paper. We want to clarify them below:
>
> Recent studies have empirically shown that LLMs possess strong capabilities in graph reasoning tasks [1, 2, 3]. At the same time, it is important to understand the underlying mechanisms behind this graph reasoning ability with the textual data [4, 5]. We focus on the substructure extraction task, motivated by prior work [2, 3] showing that LLMs can identify and extract specific substructures from a given graph.
>
> To bridge the gap, the goal of our paper is to provide **insights into why LLMs can extract substructures from text descriptions of graphs**. As clearly stated in the abstract **How can a decoder-only Transformer architecture understand underlying graph structures?**. We do not aim to argue using transformers to replace GNNs or graph transformers, nor do we propose new algorithms to solve graph tasks.
>
> To this end, we train **decoder-only** transformers using a **text-based input** format, following the GPT-2 training pipeline, to explore the mechanisms underlying their graph reasoning abilities. Therefore, connecting and comparing with graph-specific encode models such as GNNs and graph transformers are completely beyond the scope of this work.
>
> Although it is off topic of our paper, we'd like to introduce the differences between decoder-only transformer and encoder based GNN, and Graph Transformers which have been discussed in the existing papers [5, 6, 7]. GNNs and graph transformers in substructure extractions are models for graph encoding, and the output only limited in the substructure counting, instead of in indefinite length of textual description of substructure. Therefore, the mechanism is totally different as summarized in the Table:
>
> |                                | **GNNs**                                                                                    | **Graph Transformer**                                                      | **Decoder-Only Transformer (LLMs)**                         |
> | ------------------------------ | ------------------------------------------------------------------------------------------- | -------------------------------------------------------------------------- | ----------------------------------------------------------- |
> | **Input**                      | Discrete graph data (node features, adjacency matrix)                                       | Discrete graph data (node features, adjacency matrix)                      | Textual description of a graph                              |
> | **Output**                     | Scalar value (e.g., count) or fixed-size vector (e.g., for classification)                  | Scalar value (e.g., count) or fixed-size vector (e.g., for classification) | Text tokens forming a human-readable structural description in indefinite length|
> | **Learning Formulation**       | Encode graph via message passing and neighborhood aggregation; supervised on scalar outputs | Encode graph via graph-aware self-attention; supervised on scalar outputs  | Next-token prediction over graph-structured text            |
> | **Mechanism** | 1-WL [5]                                                                                       | k-WL [6]/ SEG-WL [7]                                                              | ISF (Ours)                                                        |
>
> - W: It is unclear how well the proposed methods (ISF and Tins) generalize to real-world graphs with larger size and noise.
>
> A: As claried in the previous question, the focus of this work is "How can a decoder-only Transformer architecture understand underlying graph structures?" Therefore, our work does not propose new methods for solving graph tasks. Instead, we focus on providing both theoretical and empirical foundations for why decoder-only transformers can understand structural information from textual graph descriptions. We summarize the underlying mechanisms and refer to them as ISF and Tins.
>
> We follow the data and settings used in prior work [4, 5], which showed that simplified datasets can capture key properties of decoder-only transformers, which are also reflected in the performance of LLMs. Based on this, we use simplified textual graph data to design experiments that explore how decoder-only transformers understand substructures. The findings from these foundational experiments also support and help explain the behaviors observed in LLMs, as discussed in Section 4.
>
> Meanwhile, we use molecular graphs as a special case of attributed graphs to demonstrate the theoretical foundations in real-world applications, where LLMs are capable of reasoning over attributed graphs [2, 3, 7, 10].
>
> - Q: sensitive to the format
>
> A: Thank you for your great question. This is exactly one of the key goals of our paper. Prior work has shown that LLMs are sensitive to the formulation of input structure prompts, and different formats can lead to varying performance [3, 9]. In our paper, we explain this phenomenon by comparing the EL and AL formulations. While both formats are valid for graph tasks, they differ in input length, which is an important factor, as LLMs are sensitive to sequence length and effect performance.
>
> [1] Wang H, Feng S, He T, et al. Can language models solve graph problems in natural language?[J]. Advances in Neural Information Processing Systems, 2023, 36: 30840-30861.
>
> [2] Tang J, Zhang Q, Li Y, et al. GraphArena: Evaluating and Exploring Large Language Models on Graph Computation[C]//The Thirteenth International Conference on Learning Representations.
>
> [3] Dai X, Qu H, Shen Y, et al. How Do Large Language Models Understand Graph Patterns? A Benchmark for Graph Pattern Comprehension[C]//The Thirteenth International Conference on Learning Representations.
>
> [4] Saparov A, Pawar S A, Pimpalgaonkar S, et al. Transformers Struggle to Learn to Search[C]//The Thirteenth International Conference on Learning Representations.
>
> [5] Wang S, Shen Y, Feng S, et al. Alpine: Unveiling the planning capability of autoregressive learning in language models[J]. Advances in neural information processing systems, 2024, 37: 119662-119688.
>
> [6] Chen Z, Chen L, Villar S, et al. Can graph neural networks count substructures?[J]. Advances in neural information processing systems, 2020, 33: 10383-10395.
>
> [7] Zhu W, Wen T, Song G, et al. On structural expressive power of graph transformers[C]//Proceedings of the 29th ACM SIGKDD Conference on Knowledge Discovery and Data Mining. 2023: 3628-3637.
>
> [8] Müller L, Kusuma D, Bonet B, et al. Towards principled graph transformers[J]. Advances in Neural Information Processing Systems, 2024, 37: 126767-126801.
>
> [9] Fatemi B, Halcrow J, Perozzi B. Talk like a Graph: Encoding Graphs for Large Language Models[C]//The Twelfth International Conference on Learning Representations.
>
> [10] Zhang Y, Wang H, Feng S, et al. Can LLM Graph Reasoning Generalize beyond Pattern Memorization?[C]//Findings of the Association for Computational Linguistics: EMNLP 2024. 2024: 2289-2305.

---

### Official Review · Reviewer_XYAX · 2025-07-01

**Clarity:** 3
**Significance:** 3
**Originality:** 3
**Rating:** 4
**Confidence:** 4

**Summary:**

This paper investigates the internal mechanisms by which serialized Transformer models understand graph structures, focusing on how they extract substructures from textual descriptions. It finds that the model progressively integrates node information through layer-wise Induced Substructure Filtering (ISF), where the input format (e.g., adjacency lists outperforming edge lists) and prompt design significantly impact performance. The study further proposes the Substructure Thinking (Tins) framework and validates its generalizability on attributed graphs such as molecular graphs.

**Questions:**

see in weakness

**Ethical Concerns:**

["NO or VERY MINOR ethics concerns only"]

**Final Justification:**

My concerns have been largely addressed.  I would like to maintain acceptance for this paper after considering both the response and the reviews from other reviewers.

**Limitations:**

yes

**Quality:**

2

**Strengths And Weaknesses:**

Strengths:

1、The paper explores how decoder-only Transformers process graph-structured data through sequential input, which is a relevant and underexplored topic.

2、The ISF framework provides a structured way to describe substructure extraction, and the theoretical results are generally sound and clearly stated.

Weaknesses:

1、Line 90:
There is a spelling error—"Definations" should be corrected to "Definitions"; similarly, the word "Diagnoal" is misspelled in several tables.

2、Figure 2:
The legend items such as "0431" or "2431" are unclear—what these IDs represent should be clarified for better interpretability.

3、Lines 243–257:
The paper claims that AL and EL representations are theoretically equivalent, yet EL performs significantly worse in experiments. Moreover, although EL uses more tokens and might carry richer structural information, it does not show better performance—even with deeper models, EL only matches AL's results.

4、Assumption 3.4 requires substructures to be unique in the graph, but the Multi-Num task explicitly involves multiple substructures—this inconsistency needs to be clarified.

---

> ### Author Rebuttal · Authors · 2025-07-30
>
> Overall response: Thank you for your suggestions. We would like to clarify your key concerns below.
>
> - W: spelling error.
>
> A: Thanks for your careful reading and feedback. We will address these points in the final version.
>
> - W: legend items
>
> A: The legend items represent the IDs of the extracted substructures. We conduct experiments in the directed graph setting to reduce randomness in token generation. As a result, when the model outputs a substructure, it follows a consistent central ordering of node IDs. For example, "0431" indicates that the model first identifies node 0 (with out-degree 2), followed by its neighbor 4 (out-degree 1), then node 3 (a neighbor of 4), and finally node 1 (a neighbor of both 3 and 0). Similarly, in the "2431" case, the node with out-degree 2 is node 2, serving as the structural starting point. We will address these points in the final version.
>
> - W: EL Vs. AL
>
> A: EL performs worse than AL when trained for the same number of epochs, but it eventually reaches comparable performance. In our results, we selected the epoch at which AL achieves its best performance. However, we will also provide additional information indicating when EL catches up with AL, as shown in the table below:
>
> |          | Epoch |        10000 |          20000 |        30000 |          40000 |         50000 |        60000 |
> |----------|-------|-------------:|---------------:|-------------:|---------------:|--------------:|-------------:|
> | Square (4 layers)   | AL    | 0.97 ± 0.004 | 0.98  ±  0.003 | -            | -              | -             | -            |
> |          | EL    | 0.83 ± 0.078 | 0.93  ± 0.050  | 0.99 ± 0.006 | -              | -             | -            |
> | Pentagon (5 layers) | AL    | 0.69 ± 0.017 | 0.73  ±  0.058 | 0.84 ± 0.031 | 0.92  ±  0.004 | -             | -            |
> |          | EL    | 0.61 ± 0.017 | 0.60  ± 0.019  | 0.74 ± 0.018 | 0.87  ± 0.010  | 0.89 ± 0.0056 | 0.93 ± 0.044 |
>
> EL with longer input lengths requires more training epochs to achieve the same performance as AL. Although EL and AL are theoretically equivalent in their ability to represent graph structures, the longer input sequences in EL lead to less efficient learning. We will clarify this point in the revision.
>
> - W: inconsistency of Assumption 3.4.
>
> A: The unique case is used as a special setting in Section 3.1.2, referred to as the Single-Num-Single-Shape case. This assumption is not used in the subsequent sections. Assumption 3.4 applies specifically to the scenario in Section 3.1.2, where each graph contains only one target substructure, and the Transformer is trained to extract that particular substructure. We then extend our analysis to more complex settings involving multiple substructures. In the Multi-Num-Single-Shape case, the graph contains multiple instances of a specific substructure. In the Single-Num-Multi-Shape case, the graph includes diverse substructures, and a prompt is used to guide the Transformer to extract a specific one.
>
> We will clearify the definations of these three tasks in the revision as:
>
> Single-Num-Single-Shape Task:
>
> The input is directed graphs G=(V,E) (|V|=n) and G'=(V',E') (|V'|=k), where there exists a unique copy of G' in G. The Transformer is required to find the unique copy of G' in G for the input G,G'.
>
> Multi-Num-Single-Shape Task:
>
> The input is directed graphs G=(V,E) (|V|=n) and G'=(V',E') (|V'|=k), where there might exist multiple copies of G' in G. The Transformer is required to find all copies of G' in G for the input G,G'.
>
> Single-Task-Multi-Shape Task:
> The input is directed graphs G=(V,E) (|V|=n) and G'=(V',E') (|V'|$\leq$ k), where there exists a unique copy of G' in G. The Transformer is required to find the unique copy of G' in G for the input G,G'.

---

> > ### Comment · Reviewer_XYAX · 2025-08-06
> >
> > Thank you for your feedback, which has partially addressed my concerns. I will maintain my original scores.

---

> > > ### Author Response · Authors · 2025-08-06
> > >
> > > Thank you for your feedback. We are glad to have addressed some of your concerns and clarified the paper. It seems that there may still be confusion in the rebuttal. We welcome further discussions. Please feel free to let us know your additional questions or concerns. We are happy to provide further clarification.

---

### Official Review · Reviewer_CMYh · 2025-07-02

**Clarity:** 3
**Significance:** 2
**Originality:** 2
**Rating:** 3
**Confidence:** 3

**Summary:**

The paper introduces Induced Substructure Filtration to explain how decoder-only Transformers, when fed textual graph descriptions, progressively isolate tokens belonging to a queried substructure across layers. They provide some empirical results and theoretical analysis showing this dynamic in causal transformers and show it generalises to attributed graphs (e.g., molecular graphs). The work aims to explain how sequence-based LLMs perform substructure extraction.

**Questions:**

- What are the main differences between Theorems 3.3, 3.5, 3.7, and 3.8? Also, what new insights are being gained from these theorems 3.7 and 3.8?

- The result in 3.2.2. It is not surprising also because the author trains a transformer from scratch and trains using a specified synthetic dataset that helps the model learns the correlation between the sequence of tokens in the prompt and the graph input. What surprises me is when the topology-based prompt encoder fails in the diagonal case. Could the authors elaborate on the reasons for such failure?

**Ethical Concerns:**

["NO or VERY MINOR ethics concerns only"]

**Final Justification:**

The authors provided a detailed rebuttal that addresses some of my concerns. I agree that the research question regarding the ability of LLMs on graph reasoning tasks is an interesting research question. Understanding why this capability emerges from large-scale, noisy pretraining and enables zero-shot or in-context graph reasoning is indeed very important for the field. However, the fact that the authors restrict the scope to showing that a decoder-only Transformer trained from scratch on well-defined input–output pairs can perform graph tasks does not seem to address the core question about LLMs. Therefore, I would like to maintain my initial assessment of the paper after the rebuttal.

**Limitations:**

yes

**Quality:**

2

**Strengths And Weaknesses:**

### Strengths
- The paper is well-written with nice figures
- The question posed in the paper is interesting and holds significant impact if properly addressed. However, it would be better if the authors could focus and elaborate on the differences between vanilla causal transformers and other transformer architectures such as GNNs and Graph Transformers.

### Weaknesses

- Results are not new, experiments are toy examples. They donot highlight the main difference between transformers and other graph models such as graph transformers.

- Most experiments are toy experiments where transformers are trained from scratch with well-defined synthetic data. However, the main question, how large-scale pretraining LLMs on textual data can enable model understanding of graph structures and solve graph-based tasks, remains unclear.

- A comparison to GNN models and Graph Transformer using the same set of data for training would be helpful.

- What is the connection between the graph transformer and the vanilla transformer? If the graph transformer is already well-known for handling graph data, should the transformer be good for well-represented data? The main difference is the positional encoder?

- The main results (Theorems 3.3, 3.5, 3.7, and 3.8) oversimplify the settings, which assume that the hidden dimensions go up to n^k. The results are not surprising under those assumptions

---

> ### Author Rebuttal · Authors · 2025-07-30
>
> Thank you for your suggestions. We would like to clarify your key concerns below
> - W: decoder-only transformers Vs. GNNS and graph transformers
>
> A:  We respectively point out potential misunderstandings in your comments regarding the motivation and the goal of our paper. We want to clarify them below:
>
> Recent studies have empirically shown that LLMs possess strong capabilities in graph reasoning tasks [1, 2, 3]. At the same time, it is important to understand the underlying mechanisms behind this graph reasoning ability with the textual data [4, 5]. We focus on the substructure extraction task, motivated by prior work [2, 3] showing that LLMs can identify and extract specific substructures from a given graph.
>
> To bridge the gap, the goal of our paper is to provide **insights into why LLMs can extract substructures from text descriptions of graphs**. As clearly stated in the abstract **How can a decoder-only Transformer architecture understand underlying graph structures?**. We do not aim to argue using transformers to replace GNNs or graph transformers, nor do we propose new algorithms to solve graph tasks.
>
> To this end, we train **decoder-only** transformers using a **text-based input** format, following the GPT-2 training pipeline, to explore the mechanisms underlying their graph reasoning abilities. Therefore, connecting and comparing with graph-specific encode models such as GNNs and graph transformers are completely beyond the scope of this work.
>
> Although it is off topic of our paper, we'd like to introduce the differences between decoder-only transformer and encoder based GNN, and Graph Transformers which have been discussed in the existing papers [5, 6, 7]. GNNs and graph transformers in substructure extractions are models for graph encoding, and the output only limited in the substructure counting, instead of in indefinite length of textual description of substructure. Therefore, the mechanism is totally different as summarized in the Table:
>
> |                                | **GNNs**                                                                                    | **Graph Transformer**                                                      | **Decoder-Only Transformer (LLMs)**                         |
> | ------------------------------ | ------------------------------------------------------------------------------------------- | -------------------------------------------------------------------------- | ----------------------------------------------------------- |
> | **Input**                      | Discrete graph data (node features, adjacency matrix)                                       | Discrete graph data (node features, adjacency matrix)                      | Textual description of a graph                              |
> | **Output**                     | Scalar value (e.g., count) or fixed-size vector (e.g., for classification)                  | Scalar value (e.g., count) or fixed-size vector (e.g., for classification) | Text tokens forming a human-readable structural description in indefinite length|
> | **Learning Formulation**       | Encode graph via message passing and neighborhood aggregation; supervised on scalar outputs | Encode graph via graph-aware self-attention; supervised on scalar outputs  | Next-token prediction over graph-structured text            |
> | **Mechanism** | 1-WL [5]                                                                                       | k-WL [6]/ SEG-WL [7]                                                              | The proposed ISF (Ours)                                                        |
>
> - W: toy examples and how textual data can enable model understanding.
>
> A: We follow the experimental setups of prior studies [4, 5], which suggest that simplified datasets can capture key properties of decoder-only transformers that are also reflected in the performance of LLMs. In line with these studies, we use simplified textual graph data to construct both theoretical and empirical experiments aimed at exploring the mechanism of decoder-only transformers in substructure understanding tasks. The findings from these foundational experiments also support and help explain the behaviors observed in LLMs, as discussed in Section 4.
>
> - W: The result in 3.2.2. It is not surprising and explanations for the diagonal case.
>
> A: As discussed in the previous response, we use simplified settings to uncover the basic and foundational mechanisms of decoder-only transformers. Moreover, we do not assume that the hidden dimensions scale up to $n^k$. Instead, we argue that the number of layers is the most important factor enabling decoder-only transformers to understand substructures, which is the central focus of our main results. To support this, we present an additional  experiment in which we increase the hidden dimension while training a transformer to detect triangles, using 300K training samples:
>
> | layer      |     1 |     2 |     2 |     3 |
> |------------|------:|------:|------:|------:|
> | hidden size    |   768 |   192 |   384 |   192 |
> | model size | 7.10M | 0.89M | 3.55M | 1.33M |
> | ACC        |  0.37 |  0.75 |  0.76 | 0.98 |
>
> The results show that increasing hidden size alone does not improve accuracy. For instance, 2 layers with 384 hidden size perform similarly to smaller models. In contrast, 3 layers with 192 hidden size achieve 0.98 accuracy (1.33M parameters), while 1-layer (768) and 2-layer (384) models only reach 0.37 and 0.76. This suggests that substructure understanding relies more on model depth than size. We will add this in the revision.
>
> - Q: differences between Theorems 3.3, 3.5, 3.7, and 3.8
>
> A: We first establish the basic case, single-num-single-shape, in Section 3.1.2, where only one substructure is to be extracted. Theorem 3.3 provides the foundation for how substructure extraction unfolds across layers, while Theorem 3.5 shows that the Transformer identifies the correct substructure through internal computation of a subgraph isomorphism indicator, rather than relying solely on next-token prediction. In Section 3.1.3, we address a more complex setting involving multi-num-single-shape and single-num-multi-shape cases, as illustrated by Theorems 3.7 and 3.8. These results lead to a new insight: the ISF process can operate simultaneously across multiple substructures with various prompts. This understanding helps explain how LLMs are capable of extracting diverse substructures in varying quantities [3]. Furthermore, Theorems 3.7 and 3.8 provide the theoretical foundation for the Thinking-in-Substructure phenomenon discussed in Section 5. We will further clarify these points in the final version.
>
> - Q: results in 3.2.2 setting
>
> A: In the 3.2.2 setting, substructures are not explicitly specified, which is the extension of single-num-multi-shape setting. The dataset includes diverse substructures and question types, covering both terminology and topology, all trained within a single transformer. As a result, a single graph may contain both diagonal and square substructures, and only the question prompt guides the model to extract the correct one. If the transformer fails to distinguish the prompt accurately, it may extract the wrong substructure. For instance, in the Topo1 format, the topological descriptions of diagonals and squares are similar, which can lead to confusion in prompts understanding. We will clarify this in the revision.
>
> [1] Wang H, Feng S, He T, et al. Can language models solve graph problems in natural language?[J]. Advances in Neural Information Processing Systems, 2023, 36: 30840-30861.
>
> [2] Tang J, Zhang Q, Li Y, et al. GraphArena: Evaluating and Exploring Large Language Models on Graph Computation[C]//The Thirteenth International Conference on Learning Representations.
>
> [3] Dai X, Qu H, Shen Y, et al. How Do Large Language Models Understand Graph Patterns? A Benchmark for Graph Pattern Comprehension[C]//The Thirteenth International Conference on Learning Representations.
>
> [4] Saparov A, Pawar S A, Pimpalgaonkar S, et al. Transformers Struggle to Learn to Search[C]//The Thirteenth International Conference on Learning Representations.
>
> [5] Wang S, Shen Y, Feng S, et al. Alpine: Unveiling the planning capability of autoregressive learning in language models[J]. Advances in neural information processing systems, 2024, 37: 119662-119688.
>
> [6] Chen Z, Chen L, Villar S, et al. Can graph neural networks count substructures?[J]. Advances in neural information processing systems, 2020, 33: 10383-10395.
>
> [7] Zhu W, Wen T, Song G, et al. On structural expressive power of graph transformers[C]//Proceedings of the 29th ACM SIGKDD Conference on Knowledge Discovery and Data Mining. 2023: 3628-3637.
>
> [8] Müller L, Kusuma D, Bonet B, et al. Towards principled graph transformers[J]. Advances in Neural Information Processing Systems, 2024, 37: 126767-126801.

---

> > ### Comment · Reviewer_CMYh · 2025-08-05
> > **Thank you!**
> >
> > Thank the authors for the detailed response. Could you clarify this argument "we do not assume that the hidden dimensions scale up to $n^k$"? Wouldn't Theorem 3.2-3.8 require a Transformer with O(n^k) dimension?

---

> > > ### Author Response · Authors · 2025-08-05
> > >
> > > Thank you for your response. We do not assume the hidden dimension "scale up" to $n^k$. $O(n^k)$ term is a worst-case sufficient bound to ensure that every possible ordered k-node combination in an n-node graph can be represented distinctly. It specifies an existence condition under which a Transformer can represent all such combinations, guaranteeing that the stated substructure extraction is theoretically feasible, as we prove for Thm 3.3 shown in appendix C.3. We set this bound to demonstrate that decoder‑only Transformers **have the capacity to extract substructures** from the given textual sequence. Empirically, we find that the number of Transformer layers corresponding to k determines their substructure extraction capability: when a Transformer has at least k layers, it can extract substructures containing k nodes.
> > >
> > > Furthermore, in Theorems 3.5–3.8, we build upon the result of Theorem 3.3, which establishes that substructure extraction is solvable within the
> > > $O(n^k )$ upper bound. For example, not only does the single‑num‑single‑shape case satisfy this solvability condition, but in the multi‑num‑single‑shape (Theorem 3.7) and single‑num‑multi‑shape (Theorem 3.8) cases, we also show that the upper bound depends on the largest k, and these tasks are solvable.
> > >
> > > Moreover, Section 5 demonstrates that, in practice, the extraction process does not always require reaching the theoretical upper bound of $O(n^k)$. The model only needs $O(n^q), q<k$, since complex substructures can be decomposed into simpler components, making them easier to solve. We will clarify these in the revision.

---

> > > > ### Author Response · Authors · 2025-08-08
> > > >
> > > > Dear reviewer, thank you for your feedback. We hope we've addressed all of your concerns. As we approach the end of the discussion period, we still welcome further discussions. Please feel free to let us know if you have any additional questions or concerns.

---

> > > > > ### Comment · Reviewer_CMYh · 2025-08-08
> > > > > **Thanks!**
> > > > >
> > > > > Thanks to the authors for further clarification. However, I still think the upper bounds are too loose and not practical for understanding the expressive power of the decoder-only Transformer when addressing graph tasks. These upper bounds seem trivial to me. Therefore, I would like to maintain my initial scores.

---

> > > > > > ### Author Response · Authors · 2025-08-09
> > > > > >
> > > > > > Dear Reviewer CMYh,
> > > > > >
> > > > > > Thank you again for your thoughtful and continued engagement with our work. We greatly appreciate your insights, which have helped us identify areas that could benefit from further clarification.
> > > > > >
> > > > > > We agree that the upper bound used in our paper may not be optimal. However, our primary goal in this work is to investigate why a decoder-only Transformer can perform subgraph extraction when the input graph is represented as sequences. To support this, we included a finite upper bound as a theoretical basis, rather than aiming to determine the optimal dimensionality.
> > > > > >
> > > > > > With this upper bound, we can explain why decoder-only transformers are able to extract substructures even when the input formulations vary or when dealing with complex graphs.
> > > > > >
> > > > > > We recognize that identifying tighter or optimal bounds is an interesting direction, but it falls outside the intended scope of this paper. We will make this clearer in the revised version to avoid potential confusion.
> > > > > > Thank you again for your valuable feedback.

---

### Comment · Area_Chair_mG3N · 2025-08-05

Dear reviewers,

Thank you for your effort in the reviews.

As the discussion period ends soon, please read authors' rebuttals and check if they have addressed your concern.

If authors have resolved your questions, do tell them so.

If authors have not resolved your questions, do tell them so too.

Thanks.

AC

---

### Note · Authors · 2025-08-11

Dear Chairs and Reviewers,

Thank you for your effort in organizing the review process.

Our work aims to provide insights into why LLMs can extract substructures from graph descriptions expressed in textual form. We begin by using decoder-only transformers to analyze the mechanisms that enable them to identify substructures, whether in single or multiple shapes and numbers, and even complex graphs. In addition, we examine the effects of input formulations and question prompts on their performance.

During the rebuttal phase, the main concern raised (by Reviewers CMYh, ZFQo, and J4XB) was that we did not include GNNs or Graph Transformers in our comparisons. However, we successfully clarified our contribution, which was acknowledged by Reviewer J4XB, who raised his score.

Additionally, Reviewers XYAX and CMYh pointed out that the theorems in Section 3 were described unclearly. We plan to rename them and provide new definitions to make them more precise and understandable.

We believe our work offers novel insights into why LLMs can extract substructures from textual graph descriptions. We will revise the paper to present our findings more clearly.

---

### Decision · Program_Chairs · 2025-09-17

**Decision:**

Accept (poster)

**Comment:**

The paper studies a fundamental question: How can a decoder-only Transformer architecture understand graph structures? The authors first investigate the underlying mechanism inside transformers with the substructure extraction task. Specifically, the authors formulate Induced Substructure Filtration (ISF), and validate it in LLMs. The authors further explore the capabilities of Transformers in handling diverse graph types, including attributed graphs such as molecular graphs.

Strengths:
- The paper investigates an interesting and important question.
- The ISF provides a mathematical framework for the understanding of Transformers in identifying substructures.
- The paper has a good balance of both theoretical and empirical results.

Weaknesses:
- Experiments are mostly on simplified synthetic toy data or small molecular graphs, raising questions if the results of the paper can be extended to larger graphs.
- Lacking discussions on how decoder-only transfers differ from existing GNNs or graph Transformers on the substructure extraction task.
- The results in the paper based on decoder only transformers pre-trained from scratch are disconnected with the big question: why LLMs can extract substructures from text descriptions of graphs?

Overall, the paper has academic significance in providing a new perspective on the capabilities of LLMs in extracting graph structures. While the execution relied on simplified settings/models, I do not view these as fundamental issues and they still provide useful insights to advance the field. In my opinion, the merits outweigh the shortcomings.